# Latent Diffusion Pretraining for Crystal Property Prediction

**Shrimon Mukherjee** [1] [*]   **Kishalay Das** [2] [*]   **Partha Basuchowdhuri** [1]   **Pawan Goyal** [2]   **Niloy Ganguly** [2]

## Abstract

Fast and accurate prediction of crystal properties is a central challenge in new materials design. Graph neural networks and Transformer-based models have emerged as powerful tools for this task due to their ability to encode the local structural environment of atoms within a crystal. However, these models are data-hungry and in practice labeled data for crystal properties are very scarce. Pretraining–finetuning strategies, particularly those based on diffusion models, have shown promise in addressing these limitations. In this work, we introduce a novel latent-diffusion based pretraining framework CrysLDNet, designed to mitigate the data scarcity. Our approach integrates a Variational Autoencoder (VAE) with a diffusion model during the pretraining stage. The VAE encoder maps 3D crystal structures into a smooth latent space, within which the diffusion process is applied. This latent diffusion pretraining enables the graph encoder to effectively capture structural and chemical semantics from large-scale unlabeled data, which can then be finetuned for specific property prediction tasks. Comprehensive experiments on popular DFT datasets for property prediction reveal that CrysLDNet significantly outperforms both training-from-scratch and pretrained baselines, with improvements of ***4.26%*** and ***4.90%*** on the JARVIS and MP datasets. Additionally, the learned representations remain robust in sparse-data conditions and are expressive enough to correct DFT errors when finetuned with limited experimental data. Code is available at https://github.com/shrimonmuke0202/CrysLDNet.git.

---

[*]Equal contribution   [1]School of Mathematical & Computational Sciences, Indian Association for the Cultivation of Science, Kolkata, India [2]Department of Computer Science and Engineering, Indian Institute of Technology Kharagpur, India. Correspondence to: Shrimon Mukherjee <mcssm2164@iacs.res.in>, Kishalay Das <kishalaydas@kgpian.iitkgp.ac.in>.

*Proceedings of the 43rd International Conference on Machine Learning*, Seoul, South Korea. PMLR 306, 2026. Copyright 2026 by the author(s).

## 1. Introduction

Accurate and efficient prediction of chemical properties is a critical step in the materials design pipeline. While Density Functional Theory (DFT) (Orio et al., 2009) has long been the standard tool for estimating crystal properties, its high computational cost severely limits large-scale screening. Recent advances in machine learning have provided data-driven alternatives (Gaultois et al., 2016; Lu et al., 2018; Gómez-Bombarelli et al., 2016; Xue et al., 2016; Das, 2024) that achieve DFT-comparable accuracy at a fraction of the cost. In particular, graph neural network (GNN)–based models (Xie & Grossman, 2018; Chen et al., 2019; Louis et al., 2020; Park & Wolverton, 2020; Schmidt et al., 2021; Choudhary & DeCost, 2021), followed by transformer-based architectures (Yan et al., 2022; Liao & Smidt, 2023; Lin et al., 2023; Yan et al., 2024; Shen et al., 2025b), have progressively advanced the state of the art.

Despite their success, these supervised (train-from-scratch) models require large labeled datasets, which are scarce in materials science and unevenly distributed across properties (Table 7). Also, curating property-labeled datasets through simulations or experiments is both time- and resource-intensive. As a result, existing supervised models often exhibit suboptimal performance in low-data regimes (Das et al., 2023b; Song et al., 2024), particularly for properties that lack sufficient training data. In contrast, large collections of unlabeled crystal structures with only 3D structural information are readily available. This has motivated a line of work on self-supervised pretraining for crystals, including CrysXPP (Das et al., 2022), CrysGNN (Das et al., 2023b) and Crystal Twins (Magar et al., 2022). These methods pretrain encoders on unlabeled crystal structures and subsequently fine-tune them on small property-labeled datasets, following a pretraining–finetuning paradigm.

Recent diffusion-based approaches, such as CrysDiff (Song et al., 2024), and DPF (Shen et al., 2025a) further enhance pretraining by reconstructing perturbed crystal structures. However, they operate directly in the high-dimensional feature space, jointly modeling atom types, fractional coordinates, and lattice parameters. This space is inherently heterogeneous: atom types are discrete and modeled via discrete diffusion (e.g., D3PM (Austin et al., 2021)), lattice parameters are continuous and handled by DDPMs (Ho et al., 2020), while fractional coordinates are periodic and there-

fore require score-based modeling with wrapped normal distributions (Song et al., 2020). Such heterogeneous modeling necessitates complex architectures and many diffusion steps, resulting in inefficient training and representations that remain constrained by this non-smooth input space.

In this work, we adopt a latent diffusion-based pretraining framework to overcome the limitations of conventional diffusion-based pretraining. Crystal properties such as formation energy and bandgap are fundamentally governed by atomic arrangement and lattice structure. Therefore, learning compact, expressive representations that faithfully capture both composition and 3D geometry is more beneficial than modeling the raw feature space directly. Building on this insight, we propose CrysLDNet, a novel pretraining strategy based on latent diffusion that learns robust and expressive crystal representations to improve downstream property prediction. Our framework consists of two components: a Variational Autoencoder (VAE) and a Latent Diffusion Model (LDM). The VAE encoder compresses high-dimensional crystal structures into a smooth, compact latent space, while the decoder reconstructs the original structure. The LDM then learns the distribution of these latent representations by progressively noising and denoising them using a transformer-based architecture. This latent diffusion process guides the encoder to capture richer structural semantics, yielding representations that are better aligned with downstream property prediction tasks. Empirically, we show (Figure 3) that embeddings learned via latent diffusion more accurately reconstruct atomic composition and 3D structure than feature-space diffusion methods.

Finally, the pretrained encoder is fine-tuned with a lightweight property predictor using limited labeled data. Extensive experiments on widely used DFT benchmark datasets demonstrate that CrysLDNet consistently outperforms both training-from-scratch models and existing pretrained baselines across all evaluated properties. In particular, CrysLDNet achieves substantial relative error reductions (typically in the range of **4.26% – 19.34%**) compared to strong diffusion- and GNN-based baselines, with the largest gains observed in low-data regimes, where labeled samples are scarce.

Beyond performance improvements, CrysLDNet offers an important practical advantage: it is backbone-agnostic. The encoder architecture can be seamlessly replaced with more powerful future models without modifying the latent diffusion framework or retraining strategy. In particular, we observe that replacing the VAE encoder from Matformer to PDDFormer leads to an average performance improvement of **10.46%** on JARVIS and **12.39%** on the MP dataset. This design makes CrysLDNet inherently future-proof, allowing it to benefit directly from ongoing advances in crystal representation learning and transformer architectures.

## Conflict of Interest Disclosure
The authors declare that there are no conflicts of interest.

## 2. Preliminaries
### 2.1. Crystal Representation
Crystal materials can be viewed as a 3D point cloud (Fig. 4(a)) of atoms arranged in an orderly repeating pattern. For a material with $N$ atoms in its unit cell, the structure can be defined as $M = (A, X, L)$. ***Atom Type Matrix:*** $A = [\mathbf{a_1}, \mathbf{a_2}, ..., \mathbf{a_N}]^T \in \mathbb{R}^{N \times k}$, where each $\mathbf{a_i}$ is a one-hot vector denoting the atomic type of the $i^{th}$ atom, and $k$ is the maximum number of possible atom types. ***Coordinate Matrix:*** $X = [\mathbf{x_1}, \mathbf{x_2}, ..., \mathbf{x_N}]^T \in \mathbb{R}^{N \times 3}$, where $\mathbf{x_i} \in \mathbb{R}^3$ represents the 3D coordinates of the $i^{th}$ atom in the unit cell. ***Lattice Matrix:*** $L = [\mathbf{l_1}, \mathbf{l_2}, \mathbf{l_3}]^T \in \mathbb{R}^{3 \times 3}$, which specifies how the unit cell repeats itself in 3D space along directions $\mathbf{l_1}, \mathbf{l_2}$, and $\mathbf{l_3}$ to form the periodic crystal.

**Symmetry in Crystal Structure.** Crystal materials exhibit fundamental physical symmetries that any learned representation must respect. One such property is rotational invariance, which ensures that rotating the atom coordinates and lattice matrices by any orthogonal matrix $\mathbf{Q}$ results in an equivalent representation of the same material. Another key property is periodic translation invariance, meaning that translating the atom coordinates by any arbitrary vector and applying periodic wrapping does not alter the crystal structure. A formal definition of these invariance properties is provided in Appendix B.

## 3. Proposed Methodology : CrysLDNet
### 3.1. Problem Statement
The goal of the crystal property prediction task is, given any material $\mathcal{M} = (A, X, L)$, to predict a downstream target property value $y$. Our proposed framework, CrysLDNet (Figure 1), addresses this problem through latent diffusion–based pretraining followed by finetuning. Specifically, we first leverage all available *unlabeled crystal data* $\mathcal{D}_u = \{\mathcal{M}\}_i$ to pretrain CrysLDNet, enabling it to capture intrinsic structural and chemical patterns of crystal graphs. Subsequently, we utilize a training set of *property-tagged crystal data* $\mathcal{D}_p = (\mathcal{M}_i, y_i)$ to finetune the model on the target property prediction task. During finetuning the pretrained representations are further refined and optimized for the specific downstream property.

### 3.2. CrysLDNet Pretraining
The objective of the pretraining stage is to enable the model to effectively learn and capture the structural and chemical characteristics of crystal materials from a large corpus of unlabeled data. To achieve this, we introduce a latent space diffusion pretraining strategy. The framework consists of two core components: a Variational Autoencoder

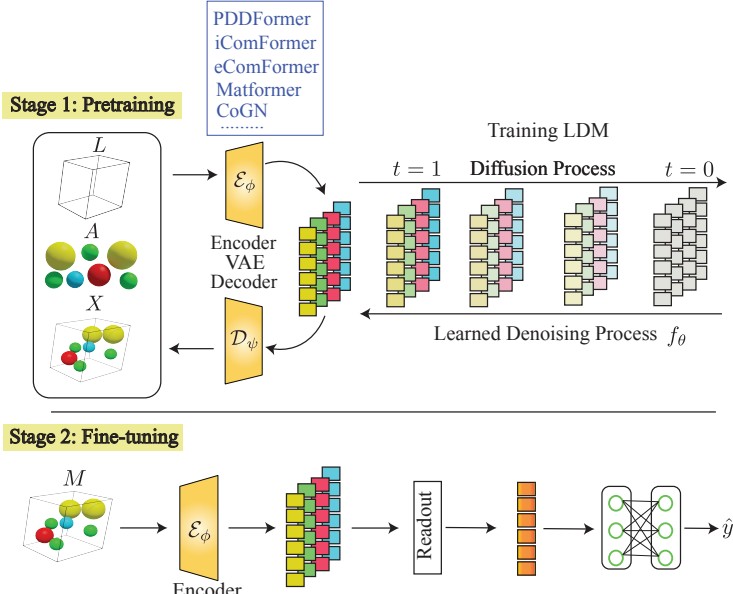

*Figure 1.* Overview of our proposed pretrain–finetune framework, CrysLDNet. In the pretraining stage, a VAE encodes crystal structures into latent representations, on which the LDM is applied to refine the encoded latent space. CrysLDNet is designed to be fully agnostic to the choice of different backbone encoder architectures. The pretrained encoder is then finetuned on labeled data for downstream tasks.

(VAE), which produces the latent representation of the crystal 3D structure and a Diffusion Model (DM), operating in a smoother, lower-dimensional latent space.

### 3.2.1. VARIATIONAL AUTOENCODER (VAE)

Our first objective is to encode the 3D crystal geometry into a lower-dimensional latent space that is both meaningful and preserves the physical symmetries inherent to crystal structures. To accomplish this, we employ a Variational Autoencoder (VAE) framework consisting of an encoder $\mathcal{E}_\phi$ and a decoder $\mathcal{D}_\psi$. The encoder module takes the crystal material $\boldsymbol{M} = (\boldsymbol{A}, \boldsymbol{X}, \boldsymbol{L})$ as input and encodes atomic types, coordinates and lattice structure into the latent space:

$$\boldsymbol{Z} = \mathcal{E}_\phi(\boldsymbol{A}, \boldsymbol{X}, \boldsymbol{L}) \tag{1}$$

where $\boldsymbol{Z} = \{\boldsymbol{z_h}\} \in \mathbb{R}^{N \times d}$, where $N$ is the number of atoms, is the set of node embeddings of the crystal materials. A key criterion in designing this encoder is that the learned latent space must preserve the physical symmetries of the crystal structure and invariance properties. Hence as encoder, we employ the PDDFormer network (Shen et al., 2025b), which ensures that the learned representations satisfy invariance to translation, rotation, and periodic transformations. Further, the decoder $\mathcal{D}_\psi$ is trained to reconstruct the 3D crystal structure from $\boldsymbol{Z}$. Specifically, we utilize three separate MLPs: one dedicated to reconstructing atomic types, another for atomic coordinates, and a third for the lattice structure from the latent representations:

$$\tilde{\boldsymbol{A}}, \tilde{\boldsymbol{X}}, \tilde{\boldsymbol{L}} = \mathcal{D}_\psi(\boldsymbol{Z}) \tag{2}$$

The whole network is trained end-to-end using a regularized reconstruction loss:

$$\mathcal{L}_{\text{VAE}} = \mathcal{L}_{\text{recon}}^{\boldsymbol{A}} + \mathcal{L}_{\text{recon}}^{\boldsymbol{X}} + \mathcal{L}_{\text{recon}}^{\boldsymbol{L}} + \alpha \mathcal{L}_{\text{reg}} \tag{3}$$

Here, $\mathcal{L}_{\text{recon}}^{\boldsymbol{A}}$, $\mathcal{L}_{\text{recon}}^{\boldsymbol{X}}$ and $\mathcal{L}_{\text{recon}}^{\boldsymbol{L}}$ represent the reconstruction losses for atom types, coordinates and lattice structure, respectively. By design, we use cross-entropy loss for $\boldsymbol{A}$ and $l_2$ loss for $\boldsymbol{X}$ and $\boldsymbol{L}$. $\mathcal{L}_{\text{reg}} = d_{\text{KL}}\{q_\phi(\boldsymbol{Z} \mid \boldsymbol{A}, \boldsymbol{X}, \boldsymbol{L}) \mid\mid p(\boldsymbol{Z})\}$ denotes KL divergence that measures how much the learned latent distribution $q_\phi(\boldsymbol{Z} \mid \boldsymbol{A}, \boldsymbol{X}, \boldsymbol{L})$ differs from the prior distribution $p(\boldsymbol{Z})$ (commonly a standard Gaussian distribution). This regularization term constrains the variance of latent embeddings, making them more stable and suitable for learning latent diffusion models.

### 3.2.2. LATENT DIFFUSION MODEL (LDM)

The VAE encoder $\mathcal{E}_\phi$ projects crystal materials into a smoother, lower-dimensional latent space. To further enhance these latent representations, we apply a diffusion model over this space. Following (Joshi et al., 2025), we adopt flow matching (Lipman et al., 2022) or Gaussian diffusion (both formulations are the same (Gao et al., 2024)), where noise is iteratively added to the latent embeddings produced by $\mathcal{E}_\phi$ over multiple time ($t$) steps to transform them toward a base distribution, and a denoising process is then employed to recover the original latent representations.

We formulate our approach by employing linear interpolation between a standard Gaussian base distribution and the target distribution defined by the VAE encoder's latent

---

**Algorithm 1** Pretraining Algorithm of CrysLDNet

---

1: **Input:** Crystal Material $M = (A, X, L)$, Property Value $y$, Encoder $\mathcal{E}_\phi$, Decoder $\mathcal{D}_\psi$, and Denoising Network $\mathcal{F}_\theta$.
2: **Stage-1: Training VAE**
3: **repeat**
4:     $Z \leftarrow \mathcal{E}_\phi(A, X, L)$
5:     $\tilde{A}, \tilde{X}, \tilde{L} \leftarrow \mathcal{D}_\psi(Z)$
6:     $\mathcal{L}^A = \text{CrossEntropyLoss}(\tilde{A}, A)$
7:     $\mathcal{L}^L = \|\tilde{L} - L\|_2^2$
8:     $\mathcal{L}^X = \|\tilde{X} - X\|_2^2$
9:     Minimize $\mathcal{L}^A + \mathcal{L}^X + \mathcal{L}^L + \lambda \mathcal{L}_{reg}$
10:     Update parameters of both $\mathcal{E}_\phi$ and $\mathcal{D}_\psi$
11: **until** Converged
12: **Stage-2: Training LDM**
13: **repeat**
14:     $Z^1 \leftarrow \mathcal{E}_\phi(A, X, L)$ // Clean Latent
15:     Sample $t \sim \mathcal{U}(0, 1)$
16:     Sample Noise $Z^0 \sim N(0, 1)^{N \times d}$
17:     $Z^t = (1 - t) Z^0 + t Z^1$
18:     $\bar{Z}^1 = \mathcal{F}_\theta(Z^t, t)$
19:     $\mathcal{L}_{LDM} = \frac{1}{(1-t)^2} \frac{1}{N} \sum_{i=0}^{N} \|z_i^1 - \bar{z}_i^1\|^2$
20:     Minimize $\mathcal{L}_{LDM}$ and Update parameters of both $\mathcal{E}_\phi$ and $\mathcal{F}_\theta$
21: **until** Converged

---

representations of 3D crystal structures. Specifically during training, given a material structure $M = (A, X, L)$, we first encode it into a lower-dimensional latent representation $Z$ using the VAE encoder $\mathcal{E}_\phi$. For convenience, we denote $Z$ as $Z^1$, representing a clean training sample at time $t = 1$. Next, we sample a noisy latent variable $Z^0$ at time $t = 0$ from a $d$-dimensional standard Gaussian distribution $\mathcal{N}(0, 1)^d$, followed by zero-centering through subtraction of its per-channel mean. Finally, we construct an interpolated noisy sample $Z^t$ at a randomly sampled time step $t \sim \mathcal{U}(0, 1)$ using linear interpolation: $Z^t = (1 - t) Z^0 + t Z^1$. Therefore, along the trajectory from the noisy latent at step $t$ to the clean latent, we define the ground-truth conditional vector field $u_t(Z^t \mid Z^1)$ as:

$$u_t(Z^t \mid Z^1) = \frac{Z^1 - Z^t}{1 - t} \tag{4}$$

By integrating this vector field $u_t(Z^t \mid Z^1)$ over time, latent samples drawn from the noisy Gaussian distribution are transformed into the true latent representations from the target distribution. We train a denoising network $\mathcal{F}_\theta$ to approximate the conditional vector field $u_t(Z^t \mid Z^1)$. The network takes as input the intermediate noisy latent $Z^t$ along with the time step $t$ and predicts the corresponding clean latent representation as: $\bar{Z}^1 = \mathcal{F}_\theta(Z^t, t)$. The denoiser is optimized by minimizing the mean squared error (MSE)

loss between the predicted conditional vector field and the ground-truth conditional vector field, formulated as:

$$\mathcal{L}_{\text{LDM}} = \frac{1}{N} \sum_{i=0}^{N} \left\| \frac{z_i^1 - z_i^t}{1 - t} - \frac{\bar{z}_i^1 - z_i^t}{1 - t} \right\|^2$$
$$= \frac{1}{(1 - t)^2} \frac{1}{N} \sum_{i=0}^{N} \|z_i^1 - \bar{z}_i^1\|^2 \tag{5}$$

As $\mathcal{F}_\theta$, we employ the Diffusion Transformer (DiT) (Peebles & Xie, 2023) architecture. During training, both the VAE encoder and the Diffusion Transformer are optimized jointly using the loss in Eq. 5. The role of the encoder is to map 3D crystal structures into latent representations, while the Diffusion Transformer is trained to predict the noise given the noisy latent input. Since the VAE encoder (PDDFormer (Shen et al., 2025b) network) has already been pretrained with the autoencoding loss in Eq. 3, it is further refined at this stage, leading to latent representations that are more enriched and expressive, which will enhance property prediction performance.

### 3.3. Backbone-Agnostic Design

A key strength of our CrysLDNet pretraining framework is that it is fully agnostic to the choice of backbone encoder architecture. The crystal graph encoder used in the VAE and the downstream property predictor can be replaced with any crystal-GNN, EGNN, or transformer architecture without modifying the rest of the pipeline. All other components, like the latent diffusion model, decoder, and task-specific heads, remain unchanged regardless of the encoder choice. This modular design enables seamless substitution of existing backbones (e.g., CGCNN, ALIGNN, DimeNet++, Matformer, Equiformer) by simply plugging them into the encoder slot. Looking ahead, any future, more powerful transformer models can be seamlessly integrated into our pretrain–finetune paradigm, and we expect them to improve performance further. (Empirical evidence at 4.3)

### 3.4. CrysLDNet Fine-tuning

During the pretraining phase, first through the VAE and subsequently via the LDM, the encoder $\mathcal{E}_\phi$ progressively captures meaningful chemical and structural semantics. Building on this, we design a property predictor tailored to specific material properties, leveraging the knowledge learned by the encoder. The property predictor is composed of the previously pretrained encoder, followed by several multilayer perceptron (MLP) layers, and is fine-tuned for downstream property prediction tasks using property labeled dataset. We begin by generating node-level representations using the encoder, as described in Eq. 1. Next, a symmetric READOUT function is applied to obtain a graph-level representation $Z_g$, ensuring invariance to node orderings. Finally, this aggregated representation is passed through an MLP,

| | | JARVIS-DFT (Choudhary et al., 2020) | | | | | | | | | Materials Project (Chen et al., 2019) | | | |
|---|---|---|---|---|---|---|---|---|---|---|---|---|---|---|
| | Model | Formation Energy | Bandgap (OPT) | Total Energy | Ehull | Bandgap (MBJ) | Bulk Modulus | Shear Modulus | SLME (%) | Spillage | Formation Energy | Bandgap (OPT) | Bulk Modulus | Shear Modulus |
| Supervised Models (Train-from-scratch) | CGCNN | 0.063 | 0.200 | 0.078 | 0.170 | 0.410 | 14.47 | 11.75 | 8.022 | 0.454 | 0.031 | 0.292 | 0.047 | 0.077 |
| | SchNet | 0.045 | 0.190 | 0.047 | 0.140 | 0.430 | 13.25 | 11.12 | 7.431 | 0.409 | 0.033 | 0.345 | 0.066 | 0.099 |
| | MEGNet | 0.047 | 0.145 | 0.058 | 0.084 | 0.340 | 14.20 | 12.25 | 7.213 | 0.445 | 0.030 | 0.307 | 0.060 | 0.099 |
| | GATGNN | 0.047 | 0.170 | 0.056 | 0.120 | 0.510 | 14.32 | 12.48 | 7.504 | 0.431 | 0.033 | 0.280 | 0.045 | 0.075 |
| | CoGN | 0.027 | 0.122 | 0.029 | 0.047 | 0.264 | 9.382 | 8.982 | 4.546 | 0.367 | 0.050 | 0.204 | 0.046 | 0.070 |
| | DimeNet++ | 0.059 | 0.239 | 0.074 | 0.142 | 0.394 | 10.50 | 10.00 | 5.291 | 0.374 | 0.049 | 0.392 | 0.041 | 0.068 |
| | Equiformer | 0.191 | 0.265 | 0.486 | 0.286 | 0.649 | 12.54 | 14.77 | 6.133 | 0.361 | 0.405 | 0.565 | 0.055 | 0.075 |
| | ALIGNN | 0.033 | 0.142 | 0.037 | 0.076 | 0.310 | 10.40 | 9.481 | 5.146 | 0.389 | 0.022 | 0.218 | 0.051 | 0.078 |
| | Matformer | 0.033 | 0.137 | 0.035 | 0.064 | 0.300 | 11.21 | 10.76 | 5.260 | 0.398 | 0.021 | 0.211 | 0.043 | 0.073 |
| | PotNet | 0.029 | 0.127 | 0.032 | 0.055 | 0.270 | 10.11 | 9.232 | 4.570 | 0.361 | 0.019 | 0.204 | 0.040 | 0.065 |
| | eComFormer | 0.028 | 0.124 | 0.032 | 0.047 | 0.282 | 10.79 | 9.826 | 4.610 | 0.373 | 0.018 | 0.202 | 0.042 | 0.073 |
| | iComFormer | 0.027 | 0.122 | 0.029 | 0.044 | 0.261 | 9.617 | 9.098 | 4.583 | 0.360 | 0.018 | 0.193 | 0.038 | 0.064 |
| | PDDFormer | 0.027 | 0.120 | 0.028 | 0.033 | 0.251 | 9.546 | 8.808 | 4.300 | 0.358 | 0.016 | 0.189 | 0.034 | 0.062 |
| Pretrain–Finetune | CrysXPP | 0.062 | 0.190 | 0.072 | 0.139 | 0.378 | 13.61 | 11.20 | 5.110 | 0.363 | 0.034 | 0.269 | 0.055 | 0.084 |
| | Crystal Twins | 0.042 | 0.160 | 0.050 | 0.132 | 0.374 | 13.41 | 11.18 | 4.967 | 0.393 | 0.034 | 0.269 | 0.051 | 0.082 |
| | CrysGNN | 0.056 | 0.183 | 0.069 | 0.130 | 0.371 | 13.42 | 11.07 | 5.452 | 0.374 | 0.033 | 0.266 | 0.043 | 0.076 |
| | CrysDiff | 0.029 | 0.131 | 0.034 | 0.062 | 0.287 | 9.875 | 9.191 | 5.030 | 0.358 | – | – | – | – |
| | DPF | 0.029 | 0.122 | 0.032 | 0.059 | 0.311 | 10.43 | 9.596 | 5.129 | 0.358 | 0.020 | 0.203 | 0.042 | 0.073 |
| | **CrysLDNet** | **0.026** | **0.118** | **0.027** | **0.032** | **0.238** | **8.817** | **8.428** | **4.120** | **0.340** | **0.015** | **0.184** | **0.032** | **0.059** |

*Table 1.* Summary of MAE results for various properties on JARVIS-DFT (left block) and Materials Project (right block). For CrysDiff on MP, results are unavailable and shown as "–". Best and second-best are in bold and underlined, respectively.

which predicts the desired material property. Formally, the property predictor can be expressed as:

$$\hat{y} = \text{MLP}_\lambda\big(\text{READOUT}\{\mathcal{E}_\phi(\mathcal{M})\}\big), \qquad (6)$$

where $\hat{y}$ is the predicted property value. The network is fine-tuned end-to-end using mean square error (MSE) objective function between predicted $\hat{y}$ and true property values $y$:

$$\min_{\phi,\lambda} \mathcal{L}_{\text{MSE}} = \|\hat{y} - y\|^2 \qquad (7)$$

By leveraging the pretrained encoder, we transfer its rich encoded knowledge into the property predictor, allowing it to benefit directly from the representations learned during the pre-training stage. During fine-tuning the pretrained representations are further refined and optimized for the specific downstream property. This significantly reduces the reliance on large-scale property-labeled datasets, enabling effective property prediction even with limited labeled data.

# 4. Experiments

## 4.1. Datasets for Pretraining and Downstream Tasks

For pretraining, we follow prior work (Shen et al., 2025a) and use 380,740 crystal structures filtered from the recent GNoME dataset (Merchant et al., 2023). Following (Shen et al., 2025a) we exclude entries that are duplicates of down-stream datasets or lack physical or chemical significance. A detailed description of the pre-trained dataset is provided in Table 6 in Appendix. Further, for the downstream crystal property prediction task, we primarily use two benchmark (property-labeled) crystal datasets: Materials Project (MP-2018.6.1) (Chen et al., 2019) and JARVIS-DFT (Choud-hary et al., 2020). Note that these are DFT benchmark datasets, where all properties are derived from DFT-based calculations. The JARVIS dataset is a widely used benchmark containing 55,722 crystal structures, and following

prior studies, we focus on nine properties: formation energy, bandgap (OPT), bandgap (MBJ), total energy, bulk modulus, shear modulus, ehull, spillage, and SLME. The Materials Project (MP) is another benchmark dataset containing 69,239 materials. From this dataset, we select four properties for evaluation: formation energy, bandgap (OPT), bulk modulus, and shear modulus. Detailed statistics of both datasets, including the number of labeled samples for each property, are reported in Table 7. Finally, to evaluate CrysLDNet's ability to mitigate DFT error bias using experimental data (Section 4.6), we use the OQMD-EXP dataset (Kirklin et al., 2015), which contains ∼1,500 crystal materials with experimentally measured formation energies.

## 4.2. Downstream Task Evaluation

**Setup.** First, we evaluate the performance of our proposed CrysLDNet on crystal property prediction tasks using aforementioned DFT benchmark datasets: JARVIS-DFT and Materials Project (MP), selecting four and nine properties, respectively. We follow prior works for dataset splits. For JARVIS, we adopt an 80 / 10 / 10 split for training, validation, and testing across all properties, consistent with Matformer (Yan et al., 2022). For MP, we follow ALIGNN (Choudhary & DeCost, 2021) for formation energy and bandgap (OPT) with 60,000 / 5,000 / 4,239 crystals as train, validation, and test, and follow Matformer for bulk and shear moduli with 4,664 / 393 / 393 crystals in the respective splits. (More details in Table 7)

**Baseline Models.** To evaluate the effectiveness of CrysLD-Net, we compare its performance with a diverse set of state-of-the-art models on crystal property prediction tasks. Specifically, we consider thirteen popular supervised models: CGCNN (Xie & Grossman, 2018), SchNet (Schütt et al., 2018), MEGNet (Chen et al., 2019),

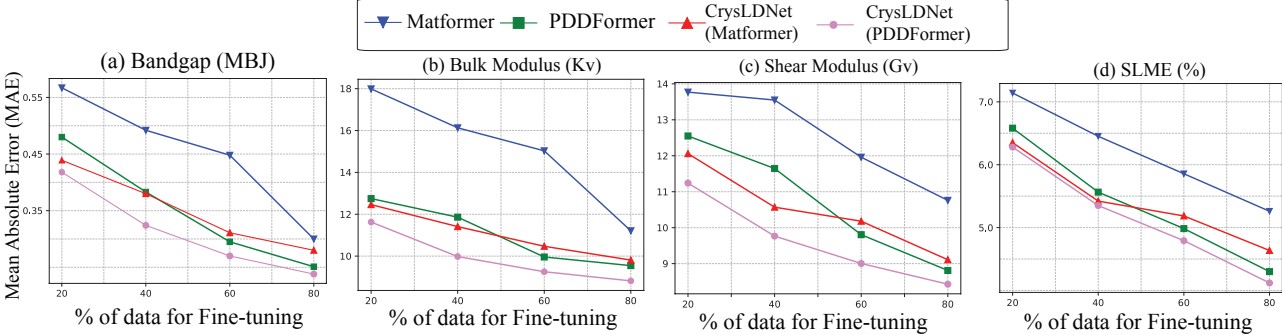

*Figure 2.* Performance comparison (MAE) under limited training data. MAE on four JARVIS properties using 20%, 40%, 60%, and 80% of finetuning data, comparing supervised baselines (PDDFormer, Matformer) with their corresponding CrysLDNet variants.

GATGNN (Louis et al., 2020), DimeNet++ (Gasteiger et al., 2020), ALIGNN (Choudhary & DeCost, 2021), Matformer (Yan et al., 2022), Equiformer (Liao & Smidt, 2023), PotNet (Lin et al., 2023), CoGN (Ruff et al., 2024), eComFormer (Yan et al., 2024), iComFormer (Yan et al., 2024), and PDDFormer (Shen et al., 2025b) all trained from scratch directly on property-labeled data. In addition, we benchmark CrysLDNet against five pretrain-finetune models for crystal materials: CrysXPP (Das et al., 2022), Crystal Twins (Magar et al., 2022), CrysGNN (Das et al., 2023b), CrysDiff (Song et al., 2024), and DPF (Shen et al., 2025a). We report the Mean Absolute Error (MAE) between predicted and ground truth values on the test set in Table 1.

**Results.** From Table 1, we observe that CrysLDNet consistently achieves superior performance (lower MAE) than all baseline models, including both supervised and pretrained approaches, across all properties and both datasets. Several key insights can be drawn from these results.

First, when compared to existing pretrain–finetune models, CrysLDNet achieves substantially larger improvements, outperforming the most competitive baseline, DPF, by **16.76%** on JARVIS and **19.34%** on MP. A potential reason for the larger improvement is that DPF uses Matformer as its encoder, which is less powerful than the PDDFormer encoder used in CrysLDNet. However, even when CrysLDNet employs Matformer as the backbone encoder (Table 2), it still achieves substantial average improvements of **7.53%** on JARVIS and **7.87%** on MP. This highlights that the performance gains primarily stem from the latent diffusion pretraining strategy rather than the choice of encoder alone. While DPF applies diffusion directly in the feature space, CrysLDNet operates in a smoother latent space, enabling it to capture underlying chemical and structural information more effectively. As a result, the learned representations are more enriched, expressive, and better suited for transfer to diverse downstream property prediction tasks.

Second, compared to the strongest supervised baseline,

PDDFormer, CrysLDNet achieves an average performance improvement of **4.26%** on JARVIS and **4.90%** on Materials Project (MP). These consistent gains over fully supervised models underscore the effectiveness of the pretrain–finetune paradigm in learning richer chemical and structural representations, which in turn leads to improved accuracy in crystal property prediction. The consistent gains over both supervised and other pretrained baselines indicate that the improvements stem primarily from the pretraining strategy itself rather than architectural modifications.

Finally, a key observation is the clear dependence of performance gains on data availability. For properties with larger training sets, such as formation energy, bandgap (OPT), total energy, and Ehull (Table 7), the improvements over PDDFormer are moderate yet consistent, with average gains of **3.0%** on JARVIS and **4.45%** on MP. In contrast, for properties with limited training data, including SLME, spillage, bulk modulus, and shear modulus (Table 7), the improvements over PDDFormer are larger, with average gains increasing to **5.27%** on JARVIS and **5.36%** on MP.

This suggests that when sufficient labeled data is available, finetuning (supervised training) already captures much of the relevant information, and pretraining provides an additional benefit. However, when we have limited training data, latent diffusion pretraining acts as a strong inductive bias, allowing the model to leverage chemical and structural patterns learned during pretraining and transfer them effectively during finetuning. To confirm this hypothesis, we do an extensive experiment on a low data regime, which is discussed in Section 4.4.

### 4.3. Backbone-Agnostic Design

A key strength of the CrysLDNet framework is its backbone-agnostic design. The crystal graph encoder used within both the VAE and the downstream property predictor can be replaced with any crystal GNN, EGNN, or transformer-based architecture without modifying the remaining components of the pipeline. To evaluate the robustness and generality

| Property | Matformer | CrysLDNet (Matformer) | PDDFormer | CrysLDNet (PDDFormer) |
|---|---|---|---|---|
| Formation Energy | 0.033 | **0.029** | 0.027 | **0.026** |
| Bandgap (OPT) | 0.137 | **0.120** | 0.120 | **0.118** |
| Total Energy | 0.035 | **0.029** | 0.028 | **0.027** |
| Ehull | 0.064 | **0.045** | 0.033 | **0.032** |
| Bandgap (MBJ) | 0.300 | **0.280** | 0.251 | **0.238** |
| Bulk Modulus (Kv) | 11.21 | **9.818** | 9.546 | **8.817** |
| Shear Modulus (Gv) | 10.76 | **9.108** | 8.808 | **8.428** |
| SLME (%) | 5.260 | **4.636** | 4.300 | **4.120** |
| Spillage | 0.398 | **0.349** | 0.358 | **0.340** |
| Formation Energy | 0.021 | **0.019** | 0.016 | **0.015** |
| Bandgap (OPT) | 0.211 | **0.188** | 0.189 | **0.184** |
| Bulk Modulus (Kv) | 0.043 | **0.038** | 0.034 | **0.032** |
| Shear Modulus (Gv) | 0.073 | **0.066** | 0.062 | **0.059** |

*Table 2*. Backbone-agnostic evaluation of CrysLDNet. MAE comparison on JARVIS (top) and MP (bottom) datasets between vanilla Matformer and PDDFormer models and their corresponding CrysLDNet variants using the same backbones.

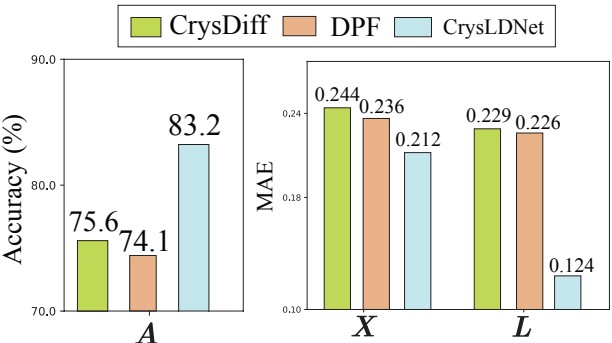

*Figure 3*. Expressiveness of pretrained representations on the GNoME dataset. Comparison of $A$, $X$ and $L$ reconstruction between CrysDiff, DPF, and CrysLDNet.

of the proposed latent pretraining framework, we instantiate CrysLDNet with two widely used crystal transformer encoders—Matformer and PDDFormer—and evaluate performance on nine properties from the JARVIS dataset and four properties from the Materials Project (MP) dataset.

The mean absolute error (MAE) results, reported in Table 2, compare the vanilla Matformer and PDDFormer models with their corresponding CrysLDNet variants that use the same backbones. Across all evaluated properties, the CrysLDNet variants consistently outperform their respective backbone models. These gains are observed for both Matformer- and PDDFormer-based encoders, indicating that the benefits of latent diffusion pretraining are not tied to a specific architectural choice. While the relative performance margins naturally decrease for more expressive encoders in well-populated data regimes, the improvements remain consistent. This trend is expected, as stronger backbones already capture a substantial portion of the underlying structure. Importantly, these results demonstrate that CrysLDNet provides complementary representational benefits and remains effective even when paired with sophisticated encoders, reinforcing its position as a future-proof pretraining framework.

This observation naturally raises the question of how much leverage CrysLDNet provides in low-data regimes, where representation learning becomes more critical role. We investigate this setting in the following section.

### 4.4. Results On Limited Training Setup
One key advantage of pretrain–finetune approaches over purely supervised models is their effectiveness in low-data settings. To evaluate this, we study the proposed latent pretraining framework under limited-data regimes by varying the amount of training data available during finetuning. Specifically, we use 20%, 40%, 60%, and 80% of the training data and evaluate performance on the same test

set. For comparison, we consider two supervised baselines, PDDFormer and Matformer, and evaluate them against their corresponding CrysLDNet variants, where PDDFormer and Matformer are used as the backbone encoders in CrysLDNet, respectively. The results, shown in Figure 2, are reported for four properties from the JARVIS dataset, which inherently contains limited training data.

First, we observe that at 20% and 40% training data, both variants of CrysLDNet consistently outperform PDDFormer and Matformer across all properties. In particular, at 40% training data, CrysLDNet achieves average improvements of **12.83%** and **22.49%** over PDDFormer and Matformer, respectively. Notably, although PDDFormer is inherently a more powerful supervised model than Matformer, CrysLDNet(Matformer) surpasses PDDFormer at 20% and 40% training data. This highlights that the knowledge acquired during pretraining is extremely beneficial in low-data settings. As the amount of labeled data increases to 60% and 80% of the full training set, the supervised models, PDDFormer and Matformer, begin to improve and achieve lower MAE. Nevertheless, the corresponding CrysLDNet variants continue to outperform their vanilla counterparts across all the properties. Overall, these results demonstrate that latent diffusion pretraining learns richer and more transferable representations, making the model more robust and effective, particularly in sparse data regimes.

### 4.5. Expressiveness of Latent Representations
We conduct an additional experiment to evaluate the expressiveness of representations learned through latent diffusion–based pretraining, in comparison with methods like CrysDiff and DPF that apply diffusion directly in feature space. For this study, we use the GNoME dataset, as it is used during pretraining, and pass each material through the pretrained encoders of CrysDiff, DPF, and CrysLDNet. We then assess their ability to reconstruct atom types ($A$), atomic coordinates ($X$), and lattice parameters ($L$). Figure 3 reports

the atom-type prediction accuracy as well as the MAE for coordinate and lattice reconstruction. Compared to CrysDiff and DPF, CrysLDNet consistently achieves stronger performance across all reconstruction tasks, with higher accuracy for $A$ and lower MAE for $X$ and $L$. These results indicate that latent diffusion–based pretraining learns more expressive representations that better capture meaningful structural information. Since crystal properties are inherently dependent on atomic configurations and lattice geometry, embeddings that better recover structural information are particularly effective for supporting downstream property prediction tasks.

| Experiment Setup | CrysGNN | CrysDiff | DPF | **CrysLDNet** |
|---|---|---|---|---|
| Finetune on DFT | | | | |
| Test on Expt | 0.253 | 0.211 | 0.217 | **0.205** |
| Finetune on DFT + 20% Expt | | | | |
| Test on 80% Expt | 0.135 | 0.102 | 0.109 | **0.097** |
| Finetune on DFT + 80% Expt | | | | |
| Test on 20% Expt | 0.096 | 0.087 | 0.070 | **0.068** |

*Table 3.* MAE on OQMD-EXP under zero-shot and 20% to 80% experimental finetuning, evaluating DFT error mitigation.

### 4.6. Results Using Experimental Data

A key challenge in materials science is the scarcity of experimental data for crystal properties (Kubaschewski et al., 1993), which limits experimental-level predictive accuracy. As a result, most approaches rely on DFT-labeled data for training; however, inherent approximations in DFT calculations introduce systematic deviations from experimental measurements, leading to DFT error bias. Prior work shows that pretraining followed by finetuning on limited experimental data can partially mitigate this bias (Das et al., 2022; 2023b). Building on this, we evaluate whether CrysLDNet further reduces DFT error when adapted to experimental data. Here, we use the OQMD-EXP dataset (Kirklin et al., 2015). All models are first trained on DFT formation energies and evaluated under three settings: zero-shot testing on the full experimental dataset, finetuning on 20% of the experimental data with testing on the remaining 80%, and finetuning on 80% with testing on the remaining 20%. We report MAE for all settings in Table 3. As experimental training data increases, all SOTA models achieve lower errors; however, CrysLDNet consistently outperforms them across all three setups, demonstrating its effectiveness in mitigating DFT error bias.

### 4.7. Influence of each component of CrysLDNet

The pretraining of CrysLDNet involves a VAE and a Latent Diffusion Model. To better understand the contribution of each component, we analyze its effects on downstream property prediction tasks. Specifically, we design two ablation experiments: (a) *VAE Only* and (b) *LDM Only*. In VAE Only case, we employ only the VAE component us-

ing the loss in Eq. 3, and then finetune the encoder on the downstream tasks. The LDM-Only model does not include a VAE; instead, it uses a PDDFormer encoder with randomly initialized parameters that produce latent representations, and the latent diffusion model operates on that. Both the encoder and the denoising network are trained jointly from scratch, and the encoder is further finetuned for downstream tasks. We report the results for four properties from JARVIS-DFT dataset in Table 4 (Top). Across both setups, we observe performance degradation for all properties, highlighting the importance of incorporating both modules during pretraining. Notably, pretraining with only the LDM achieves comparatively lower errors than only the VAE. This shows the role of diffusion models in capturing richer representations, compared to unsupervised pretraining with only the VAE. Further, in the VAE pretraining phase, we jointly reconstruct $A$, $X$ and $L$. To understand the impact of each of these reconstruction objectives, we conduct an ablation study where only subsets of atom types, coordinates, and lattice structures are reconstructed. The results, reported in Table 4 (Bottom), show performance degradation compared to CrysLDNet, indicating that reconstructing all three $(A, X, L)$ during VAE pretraining leads to better performance.

### 4.8. Analysis of the Latent Space Learned by the Encoder

We measure the Mutual Information (MI) between the learned latent representations and the underlying material structure to quantify how much structural information is preserved by the encoder (Hjelm et al., 2018). Specifically, we compute the MI $I(Z; X)$ between the latent embeddings $Z = f(x)$ and the structural attributes $X$, using it as a measure of encoder expressiveness and the extent to which structural information is retained within the latent space. We compare the VAE-only model with the full CrysLDNet framework (VAE + LDM) by evaluating MI over the entire GNoME pretraining dataset. For each crystal, latent embeddings are extracted from both models, and their MI is computed with respect to atom types and atomic coordinates. The reported values correspond to the average MI across the dataset. CrysLDNet achieves substantially higher MI for both atom types (VAE: 3.0906 $\rightarrow$ CrysLDNet: 4.5465) and atomic coordinates (VAE: 1.3124 $\rightarrow$ CrysLDNet: 2.4864), indicating that latent diffusion refinement produces richer, more expressive, and structurally grounded representations. We posit that more expressive latent representations directly translate into improved downstream property-prediction performance. This is consistent with the results in Table 4, where CrysLDNet outperforms the VAE-only baseline. To examine this effect in greater detail, we performed an ablation study in which VAE latents were progressively refined by the LDM for varying numbers of training epochs. We observed that, with increasing epochs, the LDM consistently

| Setup | Formation Energy | Bandgap (OPT) | Total Energy | Ehull | Bandgap (MBJ) | Bulk Modulus (Kv) | Shear Modulus (Gv) | SLME (%) | Spillage |
|---|---|---|---|---|---|---|---|---|---|
| VAE only | 0.031 | 0.126 | 0.032 | 0.059 | 0.284 | 10.61 | 9.773 | 4.970 | 0.374 |
| LDM only | 0.030 | 0.123 | 0.031 | 0.052 | 0.302 | 10.37 | 9.815 | 4.878 | 0.370 |
| Only A | 0.032 | 0.125 | 0.031 | 0.058 | 0.285 | 10.49 | 9.418 | 4.673 | 0.355 |
| Only X | 0.031 | 0.122 | 0.030 | 0.060 | 0.294 | 10.21 | 9.336 | 4.853 | 0.352 |
| Only L | 0.032 | 0.136 | 0.032 | 0.055 | 0.292 | 10.46 | 9.621 | 4.851 | 0.351 |
| Both A,X | 0.034 | 0.125 | 0.031 | 0.052 | 0.286 | 10.25 | 9.268 | 4.789 | 0.358 |
| Both A,L | 0.032 | 0.127 | 0.030 | 0.051 | 0.288 | 10.42 | 9.272 | 4.709 | 0.359 |
| Both L,X | 0.033 | 0.124 | 0.029 | 0.046 | 0.291 | 10.51 | 9.305 | 4.678 | 0.354 |
| **CrysLDNet** | **0.026** | **0.118** | **0.027** | **0.032** | **0.238** | **8.817** | **8.428** | **4.120** | **0.340** |

*Table 4.* Ablation studies on the JARVIS dataset, conducted to examine the impact of different pretraining components of CrysLDNet.

| Epoch | Bulk Modulus (Kv) | Shear Modulus (Gv) | SLME (%) | Spillage |
|---|---|---|---|---|
| 5 | 10.35 | 9.674 | 4.740 | 0.356 |
| 10 | 10.27 | 9.655 | 4.695 | 0.355 |
| 20 | 10.16 | 9.492 | 4.651 | 0.352 |
| 30 | 9.912 | 9.369 | 4.647 | 0.351 |
| 50 | **9.818** | **9.108** | **4.636** | **0.349** |

*Table 5.* Ablation Studies on VAE latent on epochs CrysLDNet. Here we use matformer variant of CrysLDNet.

enhanced the VAE representations, leading to monotonic improvements in property-prediction accuracy. We report the representative results on four properties from the JARVIS dataset, present in Table 5.

### 4.9. Smoothness of the Latent Space

While mutual information does not directly quantify geometric smoothness, it reflects how effectively the latent space preserves the underlying crystal structure. In practice, representations that retain meaningful structural information while suppressing noise tend to exhibit stable and consistent behavior under small input perturbations. The diffusion process further promotes such stability by learning to recover clean representations from noisy inputs, indicating the emergence of a well-behaved latent space. Therefore, mutual information serves as an indirect indicator of latent smoothness. To provide a more direct evaluation of smoothness, we additionally perform a local perturbation analysis (Rifai et al., 2011) on the full GNoME dataset. Given a perturbed input $x' = x + \epsilon$, where $\epsilon$ denotes perturbation noise and $f(\cdot) = \mathcal{E}_\phi(\cdot)$ denotes the pretrained encoder, we measure the latent response as:

$$\Delta(\sigma) = ||\text{pool}(f(x')) - \text{pool}(f(x))||_2 \quad (8)$$

where $\text{pool}(\cdot)$ denotes the graph-level readout operation that aggregates node-level latent embeddings into a crystal-level representation. In a smooth latent space, small perturbations in the input are expected to induce only small and continuous changes in the latent representation. This behavior is illustrated in Figure 7(a) and (b). Specifically, Figure 7(a) shows that the pooled latent shift increases smoothly and approximately linearly with the perturbation scale $\sigma$, indicating a continuous and well-behaved response to input

variations. Furthermore, Figure 7(b) demonstrates that the cosine similarity between the original and perturbed latent representations remains very high ($\approx 1$ for small perturbations and $> 0.97$ even at larger noise levels) and decreases gradually as $\sigma$ increases. Together, these observations indicate that the latent representations vary continuously and remain stable under perturbations, providing evidence for locally smooth latent mappings (Alain & Bengio, 2014).

### 4.10. Additional Results

Appendix E outlines joint VAE-Flow training stability of CrysLDNet. Appendix F presents the latent variance analysis, while Appendix G demonstrates the preservation of invariance during the diffusion stage. Appendix I outlines additional experiments: performance across structural complexity I.1, pretraining complexities I.2, and statistical significance test I.3, effect of reconstruction loss in stage 2 I.4, unified backbone and pretraining comparison I.5, and comparison of pretraining strategies under GNoME dataset I.6.

## 5. Conclusion

Pretrain–finetune strategies have a significant impact on crystal property prediction. In this work, we explore this direction and propose a novel latent diffusion–based pretraining framework, CrysLDNet, which learns robust and expressive crystal representations through VAE encoding and latent diffusion denoising, enabling efficient and accurate downstream property prediction. Extensive experiments on widely used DFT benchmark datasets demonstrate that CrysLDNet outperforms both training-from-scratch and existing pretrained baselines by a clear margin. We observe that the learned representations are particularly effective in sparse data regimes, leading to substantial performance improvements, and are sufficiently expressive to mitigate DFT error bias when finetuned with limited experimental data. More broadly, the proposed framework is backbone-agnostic by design, making it future-proof and allowing it to directly benefit from ongoing advances in crystal representation learning and transformer architectures.

## Impact Statement

This work introduces CrysLDNet, a pretrain-finetune framework for crystal property prediction. The proposed approach may support the generation of novel crystal materials with applications in catalysis, energy storage, and other areas of materials science and chemistry. As with any method that facilitates targeted materials discovery, there is a potential for misuse in developing materials with unintended or harmful applications; therefore, responsible use and appropriate oversight should be considered when applying such models in practice.

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

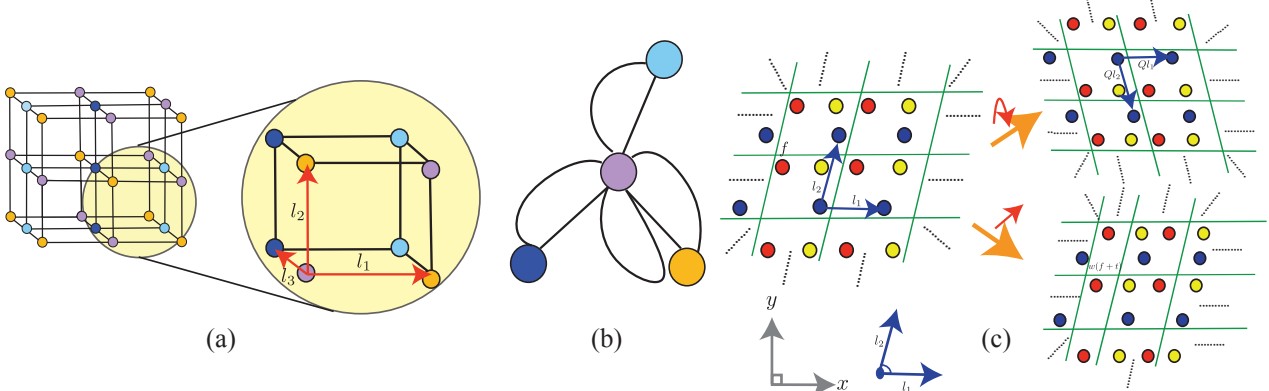

*Figure 4.* (a) A periodic crystal structure represented as a point cloud of atoms arranged in repeating patterns, along with a magnified view of a unit cell. (b) A multigraph representation of the unit cell. (c) Rotational, translational, and periodic symmetries of the crystal.

## Limitations and Future Work

CrysLDNet is pretrained on the GNoME dataset (380K crystals), which already constitutes a large-scale pretraining corpus and can naturally scale to even larger versions (e.g., 500K crystals). Since pretraining is performed only once and the learned encoder can be reused across multiple downstream tasks, the overall computational cost remains practical. Although CrysLDNet is backbone-agnostic, it depends on symmetry-preserving encoders to learn physically meaningful representations. Furthermore, the framework can be extended to other atomistic domains, such as surfaces or non-periodic systems, by adapting the VAE encoder to respect the corresponding symmetry properties, while leaving the latent diffusion framework itself unchanged.

## A. Multi-graph Construction for Crystals

Most of the state-of-the-art GNN frameworks for crystal property prediction realize a crystal material as a multi-graph structure $\mathcal{G} = (\mathcal{V}, \mathcal{E}, \mathcal{X}, \mathcal{F})$ as shown in Figure 4(b) as proposed by CGCNN (Xie & Grossman, 2018). $\mathcal{G}$ is an undirected weighted multi-graph where $\mathcal{V}$ denotes the set of nodes or atoms present in a unit cell of the crystal structure. $\mathcal{E} = \{(u, v, k_{uv})\}$ denotes a multi-set of node pairs and $k_{uv}$ denotes number of edges between a node pair $(u, v)$. $\mathcal{X} = \{(x_u | u \in \mathcal{V})\}$ denotes 92 dimensional node feature set proposed by CGCNN (Xie & Grossman, 2018). It includes different chemical properties like electronegativity, valence electron, covalent radius, etc. Finally, $\mathcal{F} = \{\{s^k\}_{(u,v)} | (u, v) \in \mathcal{E}, k \in \{1..k_{uv}\}\}$ denotes the multi-set of edge weights where $s^k$ corresponds to the $k^{th}$ bond length between a node pair $(u, v)$, which signifies the inter-atomic bond distance between two atoms.

## B. Symmetry in Crystal Structure

Crystal materials satisfy physical symmetry properties (Dresselhaus et al., 2008; Zee, 2016), one of the major challenges is the learned representation must satisfy invariance w.r.t. translation, rotation, and periodic transformations.

- ***Rotational Invariance :*** If we rotate the atom coordinates and lattice matrix, the material remains unchanged. Formally, using any orthogonal rotational matrix $\mathbf{Q} \in R^{3 \times 3}$ (satisfying $\mathbf{Q}^T \mathbf{Q} = \mathbf{I}$), if we rotate $X$ and $L$ of any material $M$ and generate new $M_R = (A, QX, QL)$, then actually different representations of the same material.

- ***Periodic Translation Invariance :*** If we translate the atom coordinates by a random vector, it will not change the structure of the material. Also, since the atoms in the unit cell can periodically repeat itself infinite times along the lattice vector, there can be many choices of unit cells and coordinate matrices representing the same material. Formally, given any material $M = (A, X, L)$, if we translate $X$ by an arbitrary translation vector $\mathbf{u} \in \mathbb{R}^3$ and if there exist a function $w[X] := X - \lfloor X \rfloor$, new generated material $M_P = (A, w(X + \mathbf{u1}^T), L)$ will be the same as $M$. Hence the structure of a crystal remains the same when applying periodic translation to $X$.

## C. Diffusion Models for Crystal Materials

Diffusion models are popular generative models that are formulated using a T-steps Markov Chain. Given a data point $\mathbf{d}_0$, the forward diffusion process gradually corrupts the data point over $T$ steps, by adding a small amount of Gaussian noise at each step:

$$q(\mathbf{d}_t|\mathbf{d}_0) = \mathcal{N}(\mathbf{d}_t|\sqrt{\bar{\alpha}_t}\mathbf{d}_0, \ (1 - \bar{\alpha}_t)\mathbf{I}) \tag{9}$$

where, $\bar{\alpha}_t = \prod_{k=1}^{t} \alpha_k$, $\alpha_t = 1 - \beta_t$ and $\{\beta_t \in (0,1)\}_{t=1}^{T}$ controls the variance of diffusion step following certain noise scheduler. Further, the reverse denoising process, which is parameterized, begins with a Gaussian noise input $\mathbf{d}_T \sim \mathcal{N}(\mathbf{0}, \mathbf{I})$ and incrementally denoises the intermediate noisy variables $\mathbf{d}_{T:1}$ to approximate the clean data $\mathbf{d}_0$ following the target data distribution:

$$p_\theta(\mathbf{d}_{t-1}|\mathbf{d}_t) = \mathcal{N}\{\mathbf{d}_{t-1} \ ; \ \mu_\theta(\mathbf{d}_{t-1}, t), \rho_t^2 \mathbf{I}\} \tag{10}$$

where $\rho_t$ is a predefined variance and mean $\mu_\theta$ is typically modeled using some neural network (U-Net for images). However, leveraging diffusion models to generate new crystal materials is challenging due to the highly multi-modal nature of their joint distribution, where each component has an independent structure and a distinct modality. On the one hand, atom types are discrete, while lattice parameters are continuous; additionally, atomic fractional coordinates are continuous but exhibit periodicity. As a result, each variable necessitates an independent diffusion framework to accommodate its unique structure.

**Diffusion on Atom Type ($A$).** Atom Type Matrix $A \in \mathbb{R}^{N \times k}$ can be considered as N discrete variables belonging to k classes and discrete diffusion model (D3PM) (Austin et al., 2021) can be leveraged for diffusion on $A$. In specific, with $\boldsymbol{a}$ as the one-hot representation of atom $a$, the transition probability for the forward process is $q(\boldsymbol{a}_t|\boldsymbol{a}_{t-1}) = Cat(\boldsymbol{a}_t; \boldsymbol{p} = \boldsymbol{a}_{t-1}\boldsymbol{Q}_t)$, where $Cat(\boldsymbol{a}; \boldsymbol{p})$ is a categorical distribution over $\boldsymbol{a}$ with probabilities $\boldsymbol{p}$ and $\boldsymbol{Q}_t$ is the Markov transition matrix at time step t, defined as $[\boldsymbol{Q}_t]_{i,j} = q(\boldsymbol{a}_t = i|\boldsymbol{a}_{t-1} = j)$. Different choices of $\boldsymbol{Q}_t$ and corresponding stationary distributions are proposed by (Austin et al., 2021) which provides flexibility to control the data corruption and denoising process.

**Diffusion on Atom Coordinates ($X$).** Coordinate Matrix $X = [\boldsymbol{x}_1, \boldsymbol{x}_2, ..., \boldsymbol{x}_N]^T \in \mathbb{R}^{N \times 3}$ contains fractional coordinates of constituent atoms, that resides in quotient space $\mathbb{R}^{N \times 3}/\mathbb{Z}^{N \times 3}$ induced by the crystal periodicity. Hence, it is not suitable to apply DDPM to model $X$, since the Gaussian distribution used in DDPM is unable to model the cyclical and bounded domain of $X$. Hence at each step of forward diffusion, noise sampled from Wrapped Normal (WN) distribution (De Bortoli et al., 2022) is added to $X$ and during denoising Score Matching Network (Song & Ermon, 2019; 2020) is leveraged to model underlying transition probability.

**Diffusion on Lattice ($L$).** Lattice Matrix $L = [l_1, l_2, l_3]^T \in \mathbb{R}^{3 \times 3}$ is a global feature of the material which determines the shape and symmetry of the unit cell structure. Since $L$ is in continuous space, we leverage the idea of the Denoising Diffusion Probabilistic Model (DDPM) (Ho et al., 2020) for diffusion on $L$.

During the reverse denoising process, a key challenge is ensuring that the learned distribution of material structures adheres to periodic E(3) invariance. To address this, existing works have utilized variants of periodic-E(3)-equivariant GNN models, such as GemNet (Gasteiger et al., 2021), or CSPNet (Jiao et al., 2023), as backbone denoising networks to guide the denoising process. Training the denoising network involves an aggregated objective function that combines cross-entropy loss, score matching loss, and $l_2$ loss for atom types, coordinates, and lattice parameters, respectively. Existing pretraining approaches (Song et al., 2024; Shen et al., 2025a) typically operate in high-dimensional feature spaces and apply diffusion-based pretraining directly on unlabeled data. Such heterogeneous modeling requires complex denoising architectures and a large number of diffusion steps to obtain high-quality crystal representations. To overcome this limitation, we propose performing pretraining in a lower-dimensional, smoother latent space, which not only enhances the learning process but also preserves essential structural and chemical information.

## D. Related Work

### D.1. Diffusion Models

The fundamental idea of the diffusion model, as initially proposed by (Sohl-Dickstein et al., 2015), is to gradually corrupt data with diffusion noise and learn a neural model to recover back data from noise. Idea of diffusion further developed in two broad categories - 1) *Score Matching Network* (Song & Ermon, 2019; 2020) and 2) *Denoising Diffusion Probabilistic Models (DDPM)* (Ho et al., 2020). In recent times diffusion models have emerged as a powerful new family of deep generative models, achieving remarkable performance records across numerous applications such as image synthesis (Dhariwal & Nichol, 2021), molecular conformer generation (Shi et al., 2021; Xu et al., 2022), molecular graph generation (Liu et al.,

2021), protein folding (Wu et al., 2021; Luo et al., 2022) etc. Recently, several studies have successfully developed latent diffusion models (LDMs) with promising results across various applications, including image generation (Vahdat et al., 2021), point clouds (Vahdat et al., 2022), and text generation (Li et al., 2022). One of the most remarkable successes among them is the Stable Diffusion (Rombach et al., 2022) models, which demonstrate surprisingly realistic text-guided image generation results.

### D.2. Crystal Property Prediction

In recent times, graph neural network(GNN) (Xie & Grossman, 2018; Chen et al., 2019; Louis et al., 2020; Park & Wolverton, 2020; Schmidt et al., 2021; Choudhary & DeCost, 2021; Yan et al., 2022; Lin et al., 2023) based approaches become popular tools for crystal property prediction. Earlier approaches (Xie & Grossman, 2018; Chen et al., 2019; Louis et al., 2020; Park & Wolverton, 2020; Schmidt et al., 2021) construct a multi-edge graph from the 3D crystal structure and apply a GNN model to encode the neighborhood structural information around an atom. Building on this, numerous studies have proposed various GNN architecture variants that integrate domain-specific knowledge into the encoder to improve crystal representation learning. ALIGNN (Choudhary & DeCost, 2021) incorporates bond angular information among edges to capture many-body interactions; whereas Matformer (Yan et al., 2022) is designed to be invariant to periodicity, enabling it to explicitly capture repeating patterns. Moreover, CrysMMNet (Das et al., 2023a) leverages multi-modal information where they fuse textual description with crystal graph structure to enhance the property prediction. One key limitation of Matformer is that it can occasionally encode crystals with distinct structures as identical graphs (Yan et al., 2024). To address this issue, ComFormer (Yan et al., 2024) proposes an SE(3)-invariant graph representation and an SO(3)-equivariant graph representation to capture both local and global geometric information, leveraging interatomic distances and atomic angular features. Then, ComFormer transforms the two types of graph representations into embeddings using transformer architectures, including node-wise and edge-wise transformer layers. This design enables ComFormer to effectively capture the geometric information of diverse structures. PDDFormer (Shen et al., 2025b) builds upon pairwise distance distribution representations to construct continuous and geometrically complete crystal graphs that remain robust to atomic position perturbations.

Data scarcity remains a significant challenge in this field, motivating the development of various graph pretraining strategies. CrysXPP (Das et al., 2022) introduced unsupervised pretraining followed by fine-tuning on property-labeled data, while CrysGNN (Das et al., 2023b) extended this idea through large-scale self-supervised pretraining, distilling knowledge from unlabeled crystal structures and transferring it to diverse property predictors to improve accuracy. Similarly, Crystal Twins (Magar et al., 2022) applied self-supervised learning to pretrain the CGCNN encoder using the Barlow Twins loss. More recently, diffusion-based pretraining frameworks have been explored for crystal structure reconstruction. CrysDiff (Song et al., 2024) employs a joint denoising diffusion model to reconstruct crystal structures from atomic compositions during pretraining, and during finetuning it conditions diffusion on target property values while keeping structures fixed. In contrast, DPF (Shen et al., 2025a) perturbs atom types, positions, and lattice constants during pretraining, reconstructs the native crystal structure, and finetunes the learned representation for downstream property prediction. CrysAtom (Mukherjee et al., 2025) derives distributed atomic representations through unsupervised learning on unlabeled crystal data, yielding substantial gains in downstream property prediction.

### D.3. Crystal Material Generation

In the past, there were limited efforts in creating novel periodic materials, with researchers concentrating on generating the atomic composition of periodic materials while largely neglecting the 3D structure. With the advancement of generative models, the majority of the research focuses on using popular generative models like VAEs or GANs to generate 3D periodic structures of materials, however, they either represent materials as three-dimensional voxel images (Court et al., 2020; Hoffmann et al., 2019; Long et al., 2021; Noh et al., 2019) and generate images to depict material structures (atom types, coordinates, and lattices), or they directly encode material structures as embedding vectors (Kim et al., 2020; Ren et al., 2022; Zhao et al., 2021). However, these models neither incorporate stability in the generated structure nor are invariant to any Euclidean and periodic transformations. Recent advancements in equivariant diffusion models have opened up a promising trajectory for the generation of novel three-dimensional periodic structures of crystal materials. CDVAE (Xie et al., 2021) was the first work that integrated a variational autoencoder (VAE) and powerful score-based decoder network, work directly with the atomic coordinates of the structures and uses an equivariant graph neural network to ensure euclidean and periodic invariance. Subsequently, numerous studies (Luo et al., 2023; Jiao et al., 2023; Zeni et al., 2023; Jiao et al., 2024; Yang et al., 2024; Das et al., 2025) have utilized diffusion models to learn the joint distribution of atom types, coordinates, and lattice structures, enabling the generation of stable periodic structures for novel materials. However, a key limitation of

these diffusion-based models is that they operate directly in high-dimensional feature spaces, jointly modeling atom types, fractional coordinates, and lattice structures. The complexity of handling these heterogeneous components demands a highly sophisticated denoising architecture and typically requires many diffusion steps to produce high-quality crystal structures. Recently, CrysLDM (Khastagir et al., 2025) and ADiT (Joshi et al., 2025) addressed this issue by introducing latent diffusion models that operate in a smoother, lower-dimensional latent space, enabling the generation of more stable and valid materials. CrysLLMGen (Khastagir et al., 2026) adopts a hybrid two-stage generation strategy, using an LLM to predict atomic compositions and a diffusion model to refine continuous structural variables significantly outperforming standalone LLM-based and denoising-based approaches across multiple benchmark tasks.

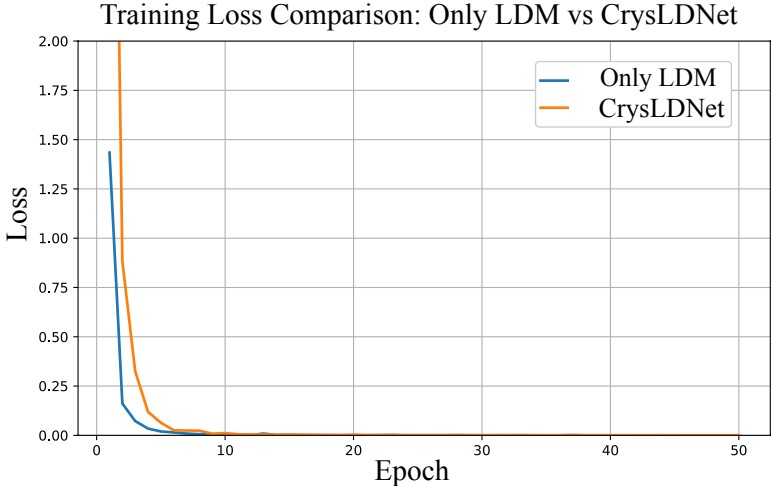

*Figure 5.* Comparison of loss curves between LDM (Without VAE Loss) and CrysLDNet.

## E. Joint VAE–Flow Training Stability

Joint VAE–Flow training can become unstable and may cause the encoder to collapse, but this usually happens when the encoder is trained from scratch with random initialization. A common and effective solution is to pretrain the encoder so that it starts from a meaningful state. In our case, the encoder is not trained from scratch; rather, we first pretrain the encoder using a VAE with a reconstruction loss over atom types, coordinates, and lattice, along with a KL regularizer (Eq. 3). Only after this step, the encoder is further refined during joint training with the LDM (Algorithm 1, Stage 2). This warm start with VAE ensures that the encoder already produces meaningful, non-collapsed latent representations, which the LDM then improves rather than driving toward a trivial constant output.

However, to investigate this phenomenon further, we compared the losses of (i) a standard LDM trained without our pretrained encoder(without VAE) and (ii) our full CrysLDNet model, and the results for 50 epochs are reported in Figure 5. In the first setting, the loss collapses almost immediately: it drops from $1.43 \rightarrow 0.16 \rightarrow 0.07 \rightarrow 0.03$ within the first four epochs, and reaches the order of $10^{-3} - 10^{-4}$ by epoch 12 e.g., 0.00148 at epoch 12 and (0.00063) at epoch 20). By epoch 50, the loss falls to $1.7 \times 10^{-4}$, indicating convergence to a near-trivial solution. Such extremely rapid loss decay is consistent with the encoder collapsing to an almost constant latent representation, allowing the flow network to minimize the matching loss trivially.

In sharp contrast, CrysLDNet does not exhibit this behavior. Its loss decreases much more gradually—from (5.39) (epoch 1) to (0.88), (0.32), and (0.11) in the first four epochs—and stabilizes in the range of $10^{-3} - 10^{-2}$ during mid-training (e.g., (0.0083) at epoch 9, (0.0039) at epoch 12, (0.0026) at epoch 17). Even at epoch 20, the loss remains at (0.00319), which is significantly higher than the collapsed LDM baseline ((0.00063)), reflecting a non-collapsed and more expressive encoder. This comparison clearly shows that collapse occurs only when the encoder lacks reconstruction or KL constraints (LDM without VAE), whereas the full CrysLDNet training remains stable and avoids the degenerate constant-latent solution the reviewer described.

## F. Latent Variance Stability of CrysLDNet

To directly verify that Stage 2 does not drive the encoder toward a trivial representation, we track the mean variance of node-level latent embeddings during Stage 2 on a fixed validation batch. As shown in Figure 6, after a brief initial transient,

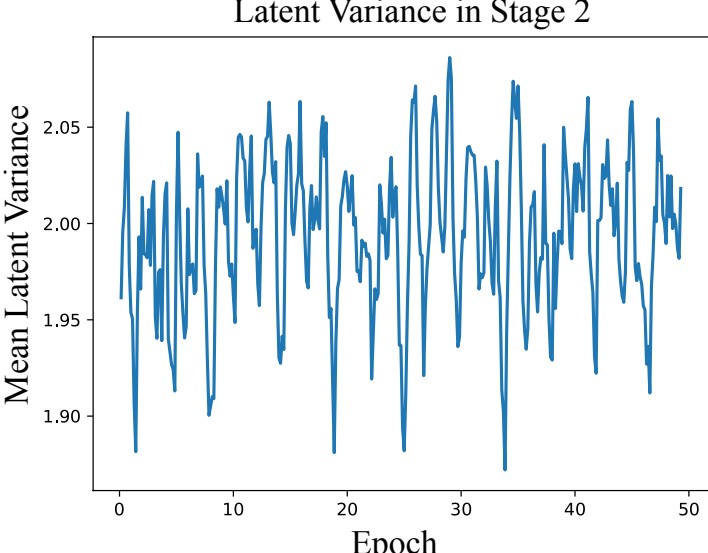

*Figure 6.* Mean latent variance during Stage 2 diffusion training. The latent variance remains stable and consistently non-zero throughout training, indicating that the latent space does not collapse and continues to preserve meaningful representation diversity.

the latent variance remains stable and clearly non-zero (approximately 1.9–2.05) across epochs, indicating that the latent space preserves meaningful expressive capacity rather than collapsing to a constant or degenerate solution.

## G. Preservation of Invariance in the Diffusion Stage

In the first stage, the use of PDDFormer as the VAE encoder already ensures rotational and translational invariance in the learned latent space. Since the DiT denoiser operates entirely on these invariant latent representations, the model design inherently preserves these properties. To further validate this, we now conduct an additional experiment where we apply rotation and translation to the same crystal and compare the denoised latent outputs. We observe high cosine similarity (0.985) and negligible differences (MAE $\approx$ 1e-6), confirming strong consistency under geometric transformations. These results verify that the latent diffusion stage preserves the invariances established in the first stage.

## H. Experimental Setup

### H.1. Implementation/ Hyperparameters details

All experiments are conducted on shared servers equipped with NVIDIA L40 GPUs. We pre-train the first stage for 50 epochs and the second stage for another 50 epochs using the AdamW optimizer with a batch size of 256, a learning rate of 1e-3, weight decay of $10^{-5}$, and a one-cycle scheduler. We then fine-tune the model on downstream crystal property prediction tasks for 1000 epochs with a batch size of 32, while keeping all other hyperparameters identical to those used during pre-training.

| Metric | $|C|$ | $|A|$ | $|T|$ | Volume | Density |
|--------|-------|-------|-------|--------|---------|
| Max    |        | 40   | 6    | 9291.69  | 24.10 |
| Min    | 380740 | 2    | 2    | 25.84    | 0.18  |
| Mean   |        | 4.10 | 4    | 436.96   | 8.34  |
| Var    |        | 0.46 | 0    | 62283.63 | 7.38  |

*Table 6.* Statistics of the GNoME dataset. Max, Min, Mean, and Var denote the maximum, minimum, average, and variance, respectively. $|A|$, $|T|$, and $|C|$ represent the number of atoms per crystal, the number of atom types per crystal, and the total number of crystal structures.

| Property | Unit | Data-size | Train/Valid/Test |
|---|---|---|---|
| Formation Energy | $eV/(atom)$ | 69239 | 60000/5000/4239 |
| Bandgap (OPT) | $eV$ | 69239 | 60000/5000/4239 |
| Bulk Modulus (Kv) | GPa | 5450 | 4664/393/393 |
| Shear Modulus (Gv) | GPa | 5450 | 4664/393/393 |
| Formation Energy | $eV/(atom)$ | 55723 | 44578/5572/5572 |
| Bandgap (OPT) | $eV$ | 55723 | 44578/5572/5572 |
| Total Energy | $eV/(atom)$ | 55723 | 44578/5572/5572 |
| Ehull | $eV$ | 55370 | 44296/5537/5537 |
| Bandgap (MBJ) | $eV$ | 18171 | 14537/1817/1817 |
| Bulk Modulus (Kv) | GPa | 19680 | 15744/1968/1968 |
| Shear Modulus (Gv) | GPa | 19680 | 15744/1968/1968 |
| SLME (%) | No unit | 9066 | 7254/906/906 |
| Spillage | No unit | 11377 | 9101/1138/1138 |

*Table 7.* Summary of different crystal properties in Materials Project (Top) and JARVIS-DFT (Bottom) datasets. Here we present total count and train/valid/test split for each properties in MP and JARVIS-DFT datasets.

### H.2. Evaluation metrics

Following prior works (Choudhary & DeCost, 2021; Xie & Grossman, 2018), we use the Mean Absolute Error (MAE) between predicted and true property values to evaluate the performance of all baseline models on the crystal property prediction task.

$$\text{MAE}(\mathcal{M}, f) = \frac{1}{m} \sum_{i=1}^{m} |f(\mathcal{M}_i) - y_i|$$

# I. Additional Results

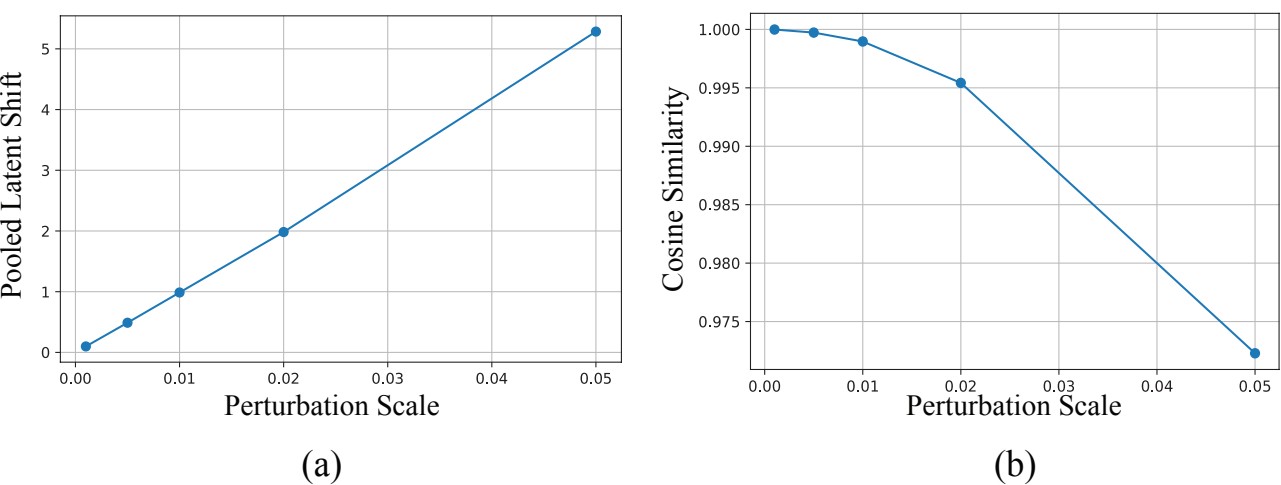

(a)                                        (b)

*Figure 7.* Local perturbation stability analysis. (a) Pooled latent shift grows smoothly with perturbation scale. (b) Cosine similarity between original and perturbed latent representations remains high even under larger perturbations, indicating a smooth and robust latent space.

### I.1. Performance Across Structural Complexity

We performed additional experiments by stratifying the training and test crystals into three structural-complexity groups according to the number of atoms ($N$) in the unit cell: simple ($N \leq 5$), moderate ($5 < N < 10$) and complex ($N > 10$). For each category, we repeat the limited-data protocol described in Section 4.4, where the same fraction of labeled samples is used for fine-tuning within the corresponding complexity bucket, and evaluation is conducted on the matching held-out test subset. This controlled setup allows us to examine whether the advantages of pretraining remain consistent across different structural regimes, rather than being averaged over the entire dataset. The corresponding results for four JARVIS properties are illustrated in Figure 8.

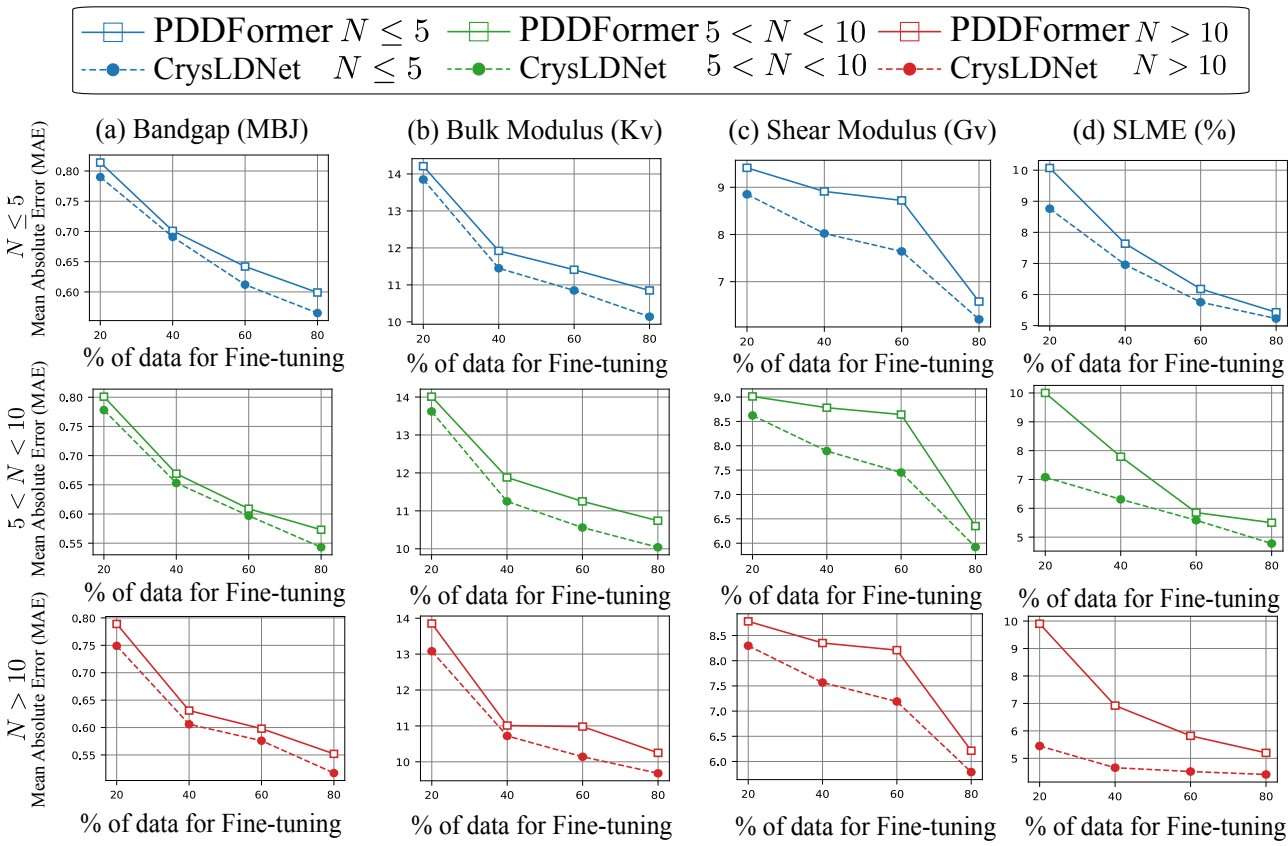

*Figure 8.* Effect of fine-tuning data size on MAE for Bandgap (MBJ), Bulk Modulus, Shear Modulus, and SLME across crystal structures of varying complexity ($N \leq 5$, $5 < N < 10$, $N > 10$) on JARVIS, comparing CrysLDNet vs PDDFormer.

The results reveal a clear monotonic increase in relative improvement, rising from 2.29% for simple structures to 5.33% for moderately complex crystals and reaching 11.10% for highly complex systems. These findings suggest that the benefits of latent diffusion pretraining become increasingly significant as structural complexity grows. In particular, the method appears especially effective for complex crystals, where capturing richer geometric and topological dependencies and higher-order interactions is crucial for accurate representation learning and downstream prediction performance.

| Parameters | DPF | VAE | LDM | **CrysLDNet** (Total) |
|---|---|---|---|---|
| Resources Used for Pre-Training | 1× NVIDIA L40 GPU server | 1× NVIDIA L40 GPU server | 1× NVIDIA L40 GPU server | 1× NVIDIA L40 GPU server |
| Memory | 15842 MB | 5986 MB | 8482 MB | 14468 MB |
| Total Training Time | ≈ 377 min | ≈ 210 min | ≈ 301 min | ≈ 511 min |
| GPU Hours (for Training) | ≈ 6.28 h | ≈ 3.5 h | ≈ 5.02 h | ≈ 8.52 h |

*Table 8.* Comparison of Computational Resources and Pre-Training Costs Across Models.

### I.2. Pretraining Complexities

Our two-stage pretraining (VAE + LDM) is slightly more expensive than a single-stage model like DPF. For clarity, we report the training times on an L40 GPU server. DPF requires ≈ 377 minutes (6.28 GPU-hours) in total, while our VAE and LDM stages take ≈ 210 minutes (3.5 GPU-hours) and 301 minutes (5.02 GPU-hours) respectively, for a total of 511 minutes (8.52 GPU-hours). We have reported more details in Table-8. Overall, this cost remains manageable in practice and is only incurred once during pretraining.

| Property | PDDFormer | CrysLDNet (PDDFormer) | | | Matformer | CrysLDNet (Matformer) | | |
|---|---|---|---|---|---|---|---|---|
| | | Mean±Std | CI | P-Value | | Mean±Std | CI | P-Value |
| Formation Energy | 0.027 | 0.026±0.0004 | (0.0255, 0.0265) | 0.005 | 0.0325 | 0.029±0.001 | (0.028, 0.030) | 0.001 |
| Bandgap | 0.120 | 0.118±0.001 | (0.117, 0.119) | 0.011 | 0.137 | 0.120±0.010 | (0.108, 0.132) | 0.019 |
| Total Energy | 0.028 | 0.027±0.0002 | (0.0268, 0.0272) | 0.0003 | 0.035 | 0.029±0.002 | (0.027, 0.031) | 0.003 |
| Ehull | 0.033 | 0.032±0.0005 | (0.031, 0.033) | 0.011 | 0.064 | 0.045±0.009 | (0.034, 0.056) | 0.010 |
| mbj Bandgap | 0.251 | 0.238±0.004 | (0.233, 0.243) | 0.007 | 0.300 | 0.280±0.010 | (0.268, 0.292) | 0.011 |
| Bulk Modulus | 9.546 | 8.817±0.240 | (8.519, 9.115) | 0.0024 | 11.21 | 9.818±1.000 | (8.576, 11.060) | 0.035 |
| Shear Modulus | 8.808 | 8.428±0.100 | (8.304, 8.552) | 0.003 | 10.76 | 9.108±1.100 | (7.742, 10.474) | 0.028 |
| SLME | 4.300 | 4.120±0.010 | (4.108, 4.132) | 0.0006 | 5.260 | 4.636±0.400 | (4.139, 5.133) | 0.025 |
| Spillage | 0.358 | 0.340±0.007 | (0.331, 0.349) | 0.004 | 0.398 | 0.349±0.010 | (0.337, 0.361) | 0.0004 |

*Table 9.* Statistical comparison of CrysLDNet across PDDformer and Matformer backbone models.

### I.3. Statistical Significance of The Results

We perform a comprehensive statistical analysis to ensure the robustness and reliability of the reported performance improvements. For each variant of CrysLDNet, we conduct five independent runs with different random seeds and report the mean, standard deviation, and 95% confidence interval (CI). In addition, for each backbone, we compute paired t-test p-values to assess the statistical significance of the improvements. Specifically, we select PDDFormer, and Matformer, along with the corresponding variants of CrysLDNet that use these encoders as backbones, to evaluate statistical significance. The complete results on JARVIS dataset are presented in Table 9.

These statistical measures demonstrate that the performance gains achieved by CrysLDNet are consistent and reproducible across multiple runs. Notably, the paired t-tests yield p-values below 0.05 for most evaluated properties, confirming that the improvements are statistically significant. Overall, this analysis verifies that the superiority of CrysLDNet does not arise from random variation but reflects genuine and meaningful performance improvements across downstream tasks.

| Setup | Formation Energy | Bandgap (OPT) | Total Energy | Ehull | Bandgap (MBJ) | Bulk Modulus (Kv) | Shear Modulus (Gv) | SLME (%) | Spillage |
|---|---|---|---|---|---|---|---|---|---|
| LDM+Decoder | 0.034 | 0.135 | 0.032 | 0.054 | 0.286 | 10.389 | 9.673 | 4.641 | 0.365 |
| **CrysLDNet** | **0.026** | **0.118** | **0.027** | **0.032** | **0.242** | **8.817** | **8.528** | **4.256** | **0.340** |

*Table 10.* Ablation study on the JARVIS dataset examining the effect of retaining reconstruction loss in Stage 2 latent diffusion training of CrysLDNet

### I.4. Effect of Reconstruction Loss in Stage 2 Latent Diffusion Training

The objectives of the two training stages are fundamentally different. In Stage 1, the VAE encoder learns a smooth and compact latent representation of 3D crystal structures through reconstruction loss, which enforces local structure-level fidelity. In Stage 2, the objective shifts toward refining these representations using latent diffusion, which promotes distribution-level consistency and encourages the latent space to evolve into a smooth and invariant manifold. Retaining reconstruction loss during this stage can introduce conflicting optimization objectives: reconstruction emphasizes exact input recovery, whereas diffusion encourages flexibility and global structure modeling. Consequently, enforcing reconstruction in Stage 2 may overly constrain the latent space and hinder the learning of more transferable and task-relevant representations. To validate this design choice, we perform an ablation study where reconstruction loss is retained during Stage 2 training (LDM+Decoder). As shown in Table 10, this variant consistently underperforms CrysLDNet across multiple JARVIS-DFT downstream tasks, demonstrating that removing reconstruction loss in Stage 2 leads to more effective representation learning and improved downstream performance.

### I.5. Unified Backbone and Pretraining Evaluation

Table 11 presents a controlled comparison in which all pretrain–finetune methods are implemented using the same PDDFormer encoder backbone and pretrained on the large-scale GNoME dataset under identical training settings. This experimental setup enables a more direct comparison of the effectiveness of different pretraining strategies while minimizing the influence of backbone architecture.

As shown in Table 11, CrysLDNet consistently outperforms prior approaches across both the JARVIS-DFT and Materials

| Property | CrysXPP | Crystal Twins | CrysGNN | DPF | CrysLDNet |
|---|---|---|---|---|---|
| Formation Energy | 0.033 | 0.030 | 0.031 | 0.029 | **0.026** |
| Bandgap (OPT) | 0.129 | 0.129 | 0.127 | 0.120 | **0.118** |
| Total Energy | 0.038 | 0.031 | 0.032 | 0.029 | **0.027** |
| Ehull | 0.039 | 0.038 | 0.037 | 0.034 | **0.032** |
| Bandgap (MBJ) | 0.266 | 0.266 | 0.262 | 0.249 | **0.242** |
| Bulk Modulus (Kv) | 9.575 | 9.281 | 9.394 | 9.248 | **8.817** |
| Shear Modulus (Gv) | 8.842 | 8.835 | 8.821 | 8.604 | **8.528** |
| SLME (%) | 4.632 | 4.602 | 4.615 | 4.534 | **4.256** |
| Spillage | 0.362 | 0.359 | 0.368 | 0.354 | **0.340** |
| Formation Energy | 0.025 | 0.025 | 0.022 | 0.017 | **0.015** |
| Bandgap (OPT) | 0.210 | 0.211 | 0.205 | 0.188 | **0.184** |
| Bulk Modulus (Kv) | 0.042 | 0.042 | 0.037 | 0.035 | **0.032** |
| Shear Modulus (Gv) | 0.065 | 0.067 | 0.063 | 0.061 | **0.059** |

*Table 11.* MAE comparison of pretrain–finetune methods on JARVIS-DFT (top) and Materials Project (bottom) benchmarks, where all methods are re-implemented using the PDDFormer encoder backbone and pretrained on the GNoME dataset under identical settings.

Project (MP) benchmarks. In particular, CrysLDNet achieves average improvements of 4.68% on JARVIS-DFT and 6.44% on MP over the second-best method, DPF. These results demonstrate that the gains achieved by CrysLDNet arise from the proposed latent diffusion pretraining framework itself, rather than solely from the choice of encoder backbone.

| Setup | Formation Energy | Bandgap (OPT) | Total Energy | Ehull | Bandgap (MBJ) | Bulk Modulus (Kv) | Shear Modulus (Gv) | SLME (%) | Spillage |
|---|---|---|---|---|---|---|---|---|---|
| CrysXPP | 0.047 | 0.177 | 0.052 | 0.129 | 0.381 | 12.902 | 10.365 | 5.121 | 0.374 |
| CrysGNN | 0.035 | 0.131 | 0.035 | 0.063 | 0.309 | 10.660 | 9.912 | 4.786 | 0.359 |
| Crystal Twins | 0.040 | 0.156 | 0.049 | 0.130 | 0.368 | 13.200 | 11.010 | 4.887 | 0.353 |
| DPF | 0.029 | 0.122 | 0.032 | 0.059 | 0.311 | 10.430 | 9.596 | 5.129 | 0.358 |
| **CrysLDNet** | **0.026** | **0.118** | **0.027** | **0.032** | **0.242** | **8.817** | **8.528** | **4.256** | **0.340** |

*Table 12.* Comparison of CrysLDNet with the pretrain-finetune models on the JARVIS dataset under a unified pretraining data setting, where all methods are pretrained on the GNoME dataset.

## I.6. Comparison of Pretraining Strategies Under a Unified GNoME Dataset Setup

To provide a controlled comparison of pretraining strategies, we additionally evaluate all pretrain–finetune baselines (CrysXPP, CrysGNN, Crystal Twins, and DPF) under a unified pretraining setup in which all methods are pretrained on the same large-scale GNoME dataset used by CrysLDNet. The corresponding results on the JARVIS-DFT benchmark are presented in Table 12. As shown in the table, CrysLDNet consistently achieves the best performance across all evaluated properties on JARVIS-DFT. These results demonstrate that, although large-scale pretraining benefits all methods, CrysLDNet is able to utilize the pretraining data more effectively, highlighting the advantage of the proposed latent diffusion–based pretraining framework beyond the impact of data scale alone.

Following prior work such as DPF, our pretraining setup leverages the large-scale unlabeled GNoME dataset to maximize the benefits of self-supervised representation learning. As described in Section 4.1, we curate approximately 380K crystal structures from GNoME after removing overlaps with downstream datasets (JARVIS and Materials Project) to avoid data leakage, and filtering out structures that do not satisfy physical or chemical validity constraints.

