# OpenReview forum: "Latent Diffusion Pretraining for Crystal Property Prediction"
_ICML.cc/2026/Conference — ICML 2026 regular_

### Official Review · Reviewer_wX5S · 2026-03-10

**Soundness:** 3
**Presentation:** 3
**Significance:** 2
**Originality:** 2
**Overall Recommendation:** 4
**Confidence:** 4

**Summary:**

The paper introduces a new method for property prediction of crystal structures called CrysLDNet, which leverages latent diffusion based self-supervised pretraining with a latent space obtained by a VAE. Section 1 introduces the challenges related to property prediction of crystal structures predicted on high compute cost of computational chemistry methods and scarce availability of training data. Here, the paper also introduces prior approaches to crystal-based property prediction including supervised method and self-supervised (i.e., reconstruction-based) diffusion models. One of the arguments made in the introduction is that the heterogenous (continuous & discrete) input space of crystal makes it difficult to learn smooth representations leading the authors to explore a latent representation. Section 2 outlines the preliminaries of crystal representation and symmetry in crystal structure.

Section 3 describes the main method of the paper with the step-wise pretraining of the two components of CrysLDNet: VAE pretraining first followed by training a latent diffusion model on the latent space of the VAE. The encoder from the VAE post diffusion is then fine-tuned for property prediction with a symmetric readout followed by MLP layers. Section 3 also details relevant loss formulation and discusses how the training framework can be applied to diverse types of encoder backbones. Section 4 describes the experiments, including relevant datasets used (filtered GNome for pretraining, JARVIS and MP for evaluation), data splits and baseline models. The results show generally better performance of CrysLDNet in the main evaluation, which is followed by further ablations and analysis. One ablation compares the application of different backbones trained with the CrysLDNet compared to supervised fine-tuning of the same backbone. Another ablation on the same backbones studies how the models' performance behave with different fractions of training data available with both supervised training and CrysLDNet showing better performance with more data included. Further experiments include an analysis of reconstruction of CrysLDNet compared to other diffusion models, CrysLDNet's performance on an experimental dataset compared to diffusion baselines and an ablation of different training losses. Section 5 provides a brief conclusion.

**Compliance With Llm Reviewing Policy:**

Affirmed.

**Final Justification:**

The authors engaged in the discussion period in sincere fashion and provided clarifications and additional details to all the feedback. This strengthen the overall clarity of the work and provides further support for the significance of the paper. As such, I am changing my overall assessment from weak reject to weak accept as after the rebuttal, in my opinion, the reasons to accept outweigh the reasons to reject.

**Key Questions For Authors:**

1. Did you try doing the pretraining with the same data as the methods you compare against (e.g., JARVIS, MP)? This would provide further evidence to the pretraining method getting better with more data. It would also be good to get more details on the considerations for the construction of the pretraining dataset.
2. For the experiment in Section 4.5, were all models trained with the same data? It would be good to disambiguate that from the test on the representations.
3. Did you try crystal generation with CrysLDNet? Why or why not?
4. Do you have evidence to showcase higher smoothness of CrysLDNet latent space compared to other diffusion methods?

**Limitations:**

The authors would benefit from discussing limitations and future work in more details. What would be needed to scale the method to larger pretraining? Is it amendable to other atomistic data (outside of crystal structures and / or crystal structures with surfaces beyond unit cell) beyond the ones currently presented? If not, how could it be adapted?

**Strengths And Weaknesses:**

The paper's strength lie mainly in its well designed experiments and clear presentation, including:
* Thorough ablations and analysis that systemically probe different aspects of the proposed method and its performance.
* Generally well-structured presentation of the method and its results.
* If successful, the proposed method could be extended and applied to broader datasets for better property prediction and other tasks. The authors can make a better case for this by discussing potential limitations in addition to outlining future directions that can build on top of CrysLDNet.

The main weaknesses relate to strengthening the significance and originality of the work:
* Latent diffusion models have been used in other domains and the paper could better motivate how their application is different from those in other fields. The adaptation to the domain of crystals could be better motivated.
* The significance of the results can be strengthened by addressing the questions below. While the current ablations probe various aspects of the method, the current draft leaves some important questions open. Being able to address those questions would give me more confidence in supporting the paper.

---

> ### Author Rebuttal · Authors · 2026-03-30
>
> **Q1. Pretraining data and dataset design**
>
> We thank the reviewer for this insightful suggestion. In the original submission, all methods—both supervised (train-from-scratch) and pretrain–finetune models—are trained and evaluated under identical JARVIS/MP protocols to ensure a fair comparison, following the settings reported in their respective papers.
>
> To take this one step further,  we now conduct controlled experiments where pretrain–finetune baselines (CrysXPP, CrysGNN, DPF and Crystal Twins) are pretrained on the **same large-scale GNoME dataset** used by CrysLDNet. Under this unified setting also, CrysLDNet consistently remains the best-performing model. This indicates that while larger data improves all methods, CrysLDNet leverages it more effectively, and the gains are not solely due to data scale but also the proposed latent diffusion–based pretraining objective. The corresponding results are provided below for JARVIS.
>
> ||Formation Energy|Bandgap|Total Energy|Ehull|
> |-|-|-|-|-|
> |CrysXPP|.047|.177|.052|.129|
> |CrysGNN|.035|.131|.035|.063|
> |Crystal Twins|.040|.156|.049| .13 |
> |DPF|.029|.122|.032|.059|
> |**CrysLDNet**|**.026**|**.118**|**.027**|**.032**|
>
> Our pretraining setup follows prior work (e.g., DPF), leveraging the large-scale unlabeled GNoME dataset to maximize pretraining benefits. As detailed in Section 4.1, we curate ~380K crystal structures from GNoME, removing overlaps with downstream datasets (JARVIS and Materials Project) to prevent data leakage, and filtering out structures that lack physical or chemical validity.
>
> **Q2. Training data clarity (Section 4.5)**
>
> Yes, all models in Section 4.5 are pretrained on the **same GNoME dataset**, ensuring a consistent data distribution across CrysDiff, DPF, and CrysLDNet. The goal is to evaluate representation expressiveness independent of dataset differences; thus, reconstruction performance (A, X, L) reflects representation quality rather than training variation. Under this setting, CrysLDNet achieves stronger reconstruction, with improvements over the second-best methods (CrysDiff/DPF) of **10.1% (A)**, **10.2% (L)**, and **2.4% (X)** (Fig. 3). These gains align with downstream improvements in Table 1, supporting the hypothesis that more expressive representations lead to better property prediction.
>
> **Q3. Crystal generation capability**
>
> Our primary objective in this work is to develop a pretraining framework for crystal property prediction, rather than a generative model for crystal structure. Accordingly, CrysLDNet is designed to learn rich and transferable representations that improve downstream prediction performance, particularly in low-data regimes. While latent diffusion models are commonly used for generation, in our approach diffusion is leveraged to refine latent representations. This results in (as shown in Section 4.5), CrysLDNet achieving  stronger reconstruction performance than diffusion-based baselines, indicating more expressive representations, which we transfer to downstream property prediction tasks. We note that unlike crystal property prediction, crystal generation with latent diffusion models is already studied with dedicated SOTA approaches, still  extending CrysLDNet for generation may be  an interesting future direction, but beyond the scope of this paper.
>
> **Q4. Evidence latent space smoothness**
>
> Yes, we provide evidence for improved smoothness of the latent space learned by CrysLDNet. Appendix G.1 analyzes the representations using **Mutual Information (MI)**, which measures how well latent variables retain structural information (e.g., atom types (A) and coordinates (X)). Higher MI indicates more consistent and less noisy representations, reflecting a smoother latent space. We observe substantially higher MI for CrysLDNet vs. VAE-only (A:  VAE: **3.09** → CrysLDNet : **4.55**;  X: VAE:  **1.31** → CrysLDNet : **2.49**), indicating richer embeddings. Consistently, Section 4.5 shows improved atom-type reconstruction and lower MAE for coordinates and lattice vs. CrysDiff/DPF, demonstrating more expressive and structurally faithful representations.
>
> **Scalability and Adaptability**
>
> CrysLDNet is pretrained on the GNoME dataset (380K crystals), which is already large-scale, and can naturally scale to larger versions (500K). Since pretraining is a one-time cost and the encoder can be reused across multiple downstream tasks, the overall computational expense remains practical. Detailed complexity analysis is provided in Table 8 and Appendix G.2.
>
> While CrysLDNet is backbone-agnostic, it relies on symmetry-preserving encoders for physically meaningful representations. The framework can be extended to other atomistic domains (e.g., surfaces or non-periodic systems), but this would require adapting the VAE encoder to match the relevant symmetry properties. The latent diffusion framework itself remains unchanged, and we will clarify this future direction in the final version.

---

> > ### Author Rebuttal · Reviewer_wX5S · 2026-04-03
> >
> > Thank you for your reply and additional details. Many of the questions related to dataset and consistency in pretraining have been addressed. The remaining questions include:
> >
> > * Can you describe more details on the mutual information study in Appendix G1? For example, what set of structures were the metrics computed on? How does mutual information provide evidence for smooth representations? While the empirical results support better prediction performance, it would be good to support the claim of smooth representations in a more detailed manner.
> > * Putting CrysLDNet in context with other latent diffusion methods in adjacent fields and motivating the application to property prediction for materials.

---

> > > ### Author Response · Authors · 2026-04-06
> > >
> > > **Q1.**
> > >
> > > The mutual information (MI) analysis evaluates how well latent representations capture crystal structure (atom types and coordinates). We compute MI $I(Z;X)$ between latent embeddings $Z=f(x)$ and structural attributes $X$ [1], measuring encoder expressiveness, i.e., how much structural information is preserved in the latent space.
> > >
> > > **Dataset and protocol**
> > >
> > > We compute MI over the full **GNoME pretraining dataset**. For each crystal, we extract latent embeddings from both the VAE-only model and the full CrysLDNet(VAE + latent diffusion), and compute their MI with respect to atom types and atomic coordinates. Reported values are averaged across the dataset.
> > >
> > > **Key findings**
> > >
> > > We observe that CrysLDNet consistently yields significantly higher MI compared to the VAE-only model: Atom types: VAE: 3.0906 → CrysLDNet: 4.5465, Atomic coordinates: VAE: 1.3124 → CrysLDNet: 2.4864. This indicates that the diffusion-based refinement captures both compositional and geometric aspects more effectively.
> > >
> > > |Metric|VAE|CrysLDNet|
> > > |---|---|---|
> > > |Atom types|3.0906|**4.5465**|
> > > |Atomic coordinates|1.3124|**2.4864**|
> > >
> > > **Relation to smoothness**
> > >
> > > While mutual information does not directly quantify geometric smoothness, it captures how well the latent space preserves underlying crystal structure. In practice, representations that retain meaningful structure and suppress noise tend to exhibit more stable and consistent behavior under small input variations. Diffusion enforces stability by recovering clean representations from noisy inputs, indicating a well-behaved latent space. Thus it captures the smoothness indirectly.
> > >
> > > To also have a more **direct evaluation of smoothness**, we now have performed a local perturbation analysis [2] using full **GeNoME dataset**. Given a perturbed input $x' = x + \epsilon$, we measure the latent response, $Δ(σ)=∥pool(f(x′))−pool(f(x))∥_{2}$.(A smooth latent space is expected to have the response $\Delta(\sigma)$ to be small. We illustrate this behavior through **Fig. [A,B]** ). Fig. A shows the pooled latent shift increases smoothly and approximately linearly with perturbation scale $\sigma$, indicating a continuous and well-behaved response to input variations. Fig. B shows that cosine similarity between original and perturbed latent representations remains high (≈1 for small perturbations and $>0.97$ even at larger noise levels) and decreases gradually as $\sigma$ increases. This shows latent representations vary continuously and remain stable under perturbations indicating locally smooth mappings [3].
> > >
> > > Fig. [A,B]: https://anonymous.4open.science/r/smoothness_latent_space-B68D/smoothness.png
> > >
> > > [1] https://arxiv.org/abs/1808.06670 , [2] https://icml.cc/2011/papers/455_icmlpaper.pdf  [3] https://arxiv.org/abs/1211.4246
> > >
> > > **Q2.**
> > >
> > > **Diffusion Models for Discriminative Tasks**:
> > > Originally developed for generation, diffusion models are now used for discriminative pretraining. Prior work in molecular property prediction [4] shows denoising objectives learn useful representations for downstream tasks. This extends across domains: protein models such as DPLM [5] use diffusion-based generative pretraining, while vision methods like ADDP [6] use a unified denoising framework for both generation and recognition. Diffusion has also been applied to domain adaptation [7], enabling improved cross-domain generalization. These works show denoising is effective for representation learning.
> > >
> > > **Latent Diffusion for Discriminative Tasks**:
> > > Recent studies explore latent diffusion for discriminative learning, motivated by improved efficiency and smoother representations. It has been applied to protein sequences [8] and domain adaptation [7], showing benefits for downstream prediction. However, latent diffusion remains underexplored compared to feature-space diffusion, especially in structured scientific domains.
> > >
> > > **Limitations of Existing Approaches**:
> > > In crystal materials, prior diffusion-based pretraining operates in input space [9], reconstructing perturbed structures. Crystal representations are high-dimensional and heterogeneous, involving discrete atom types, continuous lattice parameters, and periodic coordinates [10]. This complexity makes the input space challenging to model directly and can hinder efficient learning and representation quality.
> > >
> > > **Our Contribution**:
> > > We propose a two-stage latent diffusion pretraining framework for crystal property prediction. First, we learn a symmetry-aware latent space; then, diffusion refines representations in this space, improving smoothness, efficiency, and physical alignment. To our knowledge, this is the first application of latent diffusion pretraining in this domain.
> > >
> > > [4] https://arxiv.org/abs/2206.00133 [5] https://arxiv.org/abs/2410.13782 [6] https://arxiv.org/abs/2306.05423 [7] https://arxiv.org/abs/2510.00478 [8] https://arxiv.org/abs/2503.18551 [9] https://ojs.aaai.org/index.php/AAAI/article/view/28748 [10] https://arxiv.org/abs/2503.03965

---

### Official Review · Reviewer_tkjV · 2026-03-11

**Soundness:** 3
**Presentation:** 3
**Significance:** 3
**Originality:** 3
**Overall Recommendation:** 4
**Confidence:** 4

**Summary:**

This paper proposes CrysLDNet, which uses a VAE to encode crystal structures into a latent space and then applies a latent diffusion model to learn richer representations. After pretraining, the encoder is fine-tuned on downstream tasks. Experiments on JARVIS and Materials Project show that CrysLDNet outperforms both supervised baselines and prior pretrain-finetune methods.

**Compliance With Llm Reviewing Policy:**

Affirmed.

**Final Justification:**

This paper is well-motivated and practically relevant. It studies an important problem in crystal property prediction, where labeled data are limited and pretraining on unlabeled crystal structures is a natural and useful direction. The rebuttal addressed my main concern, and I'll keep my positive score.

**Key Questions For Authors:**

Table 1 shows that CrysLDNet performs best, and PDDFormer is the next strongest model in many cases. What would happen if other pretrain-finetune methods also used PDDFormer as the backbone?

**Limitations:**

Yes

**Strengths And Weaknesses:**

Strengths: 1) This paper is well motivated. The authors study a practical problem. Crystal property prediction often suffers from limited labeled data. Pretraining on unlabeled crystal structures is a reasonable direction. The framework is designed to be backbone-agnostic. The encoder can be replaced without changing the rest of the pipeline. This design is flexible. 2) The experimental results are strong. Table 1 reports results for both from-scratch-trained models (such as PDDFormer) and pretrain-then-finetune models (such as CrysXPP, Crystal Twins, CrysGNN, CrysDiff, and DPF). CrysLDNet shows consistent gains over prior models.

Weaknesses: The comparison with prior pretraining-finetuning methods is not deep enough. For methods such as CrysXPP, CrysGNN, Crystal Twins, CrysDiff, and DPF, Table 1 reports their performance on JARVIS-DFT and Materials Project, but does not replace their encoder backbones with a stronger one, such as PDDFormer.

---

> ### Author Rebuttal · Authors · 2026-03-31
>
> **Q1. Backbone comparison with PDDFormer**
>
> In Table 1, we follow the standard practice of using the original encoder designs and reported settings of prior pretrain–finetune methods (CrysXPP, CrysGNN, Crystal Twins, CrysDiff, and DPF) to ensure consistency with their published results. Since these methods are typically designed and tuned with specific backbones, directly modifying them requires careful reimplementation and tuning.
>
> However, we agree with the reviewer’s concern and, to address this, we have conducted additional experiments where we replace the encoder backbones with PDDFormer and pretrain all models on the same large-scale GNoME dataset used by CrysLDNet, followed by evaluation under identical settings. We note that we cannot report PDDFormer results for CrysDiff, as its fine-tuning source code is not available. The corresponding results are provided below for JARVIS and MP.
>
> | Method                         | Formation Energy | Band Gap (OPT) | Total Energy | Ehull | Bandgap (MBJ) | Bulk Modulus | Shear Modulus | SLME (%) | Spillage |
> |--------------------------------|------------------|----------------|--------------|-------|----------------|--------------|----------------|----------|----------|
> | CrysXPP            | 0.033            | 0.129          | 0.038        | 0.039 | 0.266          | 9.575        | 8.842          | 4.632    | 0.362    |
> | CrysGNN        | 0.031            | 0.127          | 0.032        | 0.037 | 0.262          | 9.394        | 8.821          | 4.615    | 0.368    |
> | Crystal Twins       | 0.030            | 0.129          | 0.031        | 0.038 | 0.266          | 9.281        | 8.835          | 4.602    | 0.359    |
> | DPF               | 0.029            | 0.120          | 0.029        | 0.034 | 0.249          | 9.248        | 8.604          | 4.534    | 0.354    |
> | **CrysLDNet**                  | **0.026**       | **0.118**      | **0.027**    | **0.032** | **0.242**   | **8.817**    | **8.528**      | **4.256** | **0.340** |
>
> | Method                         | Formation Energy | Band Gap (OPT) | Bulk Modulus | Shear Modulus |
> |--------------------------------|------------------|----------------|--------------|----------------|
> | CrysXPP           | 0.025            | 0.210          | 0.042        | 0.065          |
> | CrysGNN            | 0.022            | 0.205          | 0.037        | 0.063          |
> | Crystal Twins      | 0.025            | 0.211          | 0.042        | 0.067          |
> | DPF        | 0.017            | 0.188          | 0.035        | 0.061          |
> | **CrysLDNet**                  | **0.015**        | **0.184**      | **0.032**    | **0.059**      |
>
> From the tables, we observe that with the stronger PDDFormer backbone, CrysLDNet achieves average improvements of **4.68% on JARVIS** and **6.44% on MP** over the second-best method, DPF.  This analysis allows us to more clearly separate the impact of the pretraining strategy from the choice of backbone, and emphatically shows  the effectiveness and novelty of our proposed pre-training method. We thank the reviewer for seeking this clarification.

---

> > ### Author Rebuttal · Reviewer_tkjV · 2026-04-02
> >
> > Thank you to the authors for the thoughtful rebuttal. I do not have further questions at this stage. I appreciate the clarifications provided by the authors. While the rebuttal was helpful, my overall evaluation remains the same, and I therefore intend to keep my score unchanged.

---

> > > ### Author Response · Authors · 2026-04-07
> > >
> > > We thank the reviewer for their time and thoughtful engagement with our work. We are glad that the rebuttal helped clarify our contributions and that we were able to address all the questions raised.
> > >
> > > We would also like to highlight the additional experiments and analyses conducted during the rebuttal phase in response to other reviewers:
> > >
> > > - We conducted a structural complexity analysis by stratifying crystals based on the number of atoms in the unit cell. We observe that the relative improvement of CrysLDNet increases from 2.29% (simple) to 11.10% (complex), indicating stronger gains on structurally complex crystals (cc7e).
> > > - We provided Mutual Information (MI) analysis to quantify representation quality. CrysLDNet shows substantial improvements over VAE-only indicating richer and more informative latent representations (wX5S).
> > > - To directly evaluate smoothness, we introduced a local perturbation analysis, showing that latent representations vary smoothly under input perturbations, with gradual changes and high cosine similarity, confirming a well-behaved latent space (wX5S).
> > > - We performed an invariance preservation experiment, demonstrating that the latent diffusion stage maintains symmetry properties, with cosine similarity of ≈0.985 under rotation and translation transformations (cc7e).
> > > - To address concerns about collapse in Stage 2, we conducted a latent variance analysis, showing that the variance remains stable and non-zero (~1.9–2.05) throughout training, confirming that the latent space retains expressive capacity (cc7e).
> > > - We added an ablation study on Stage-2 objectives, showing that retaining reconstruction loss (LDM+Decoder) degrades performance compared to CrysLDNet, validating our design choice of separating reconstruction and diffusion objectives (cc7e).
> > >
> > > We believe the revisions and additional experiments significantly strengthen the work. We would greatly appreciate a reconsideration of the score based on these updates.
> > >
> > > We look forward to your response.
> > >
> > > Thank you,
> > >
> > > The Authors

---

### Official Review · Reviewer_cc7e · 2026-03-12

**Soundness:** 3
**Presentation:** 3
**Significance:** 3
**Originality:** 3
**Overall Recommendation:** 4
**Confidence:** 2

**Summary:**

This paper proposes CrysLDNet, a latent diffusion-based pretraining framework for crystal property prediction. By combining a VAE and an LDM, it performs diffusion and training in a low-dimensional space to learn the structural and chemical representations of crystal materials. Under various data conditions, its performance consistently outperforms that of the baseline models.

**Compliance With Llm Reviewing Policy:**

Affirmed.

**Final Justification:**

I thank the authors for their detailed response and the extensive additional experiments. The new results have largely resolved my concerns. While the experimental evidence is quite comprehensive, I believe that a further theoretical proof for the VAE and diffusion integration would better substantiate the framework's overall rationality.

**Key Questions For Authors:**

- Can you further explain the training methods of the pretrained baseline models, and clarify whether these models were trained under the same conditions?
- In the first stage, this paper uses PDDFormer as the VAE encoder to guarantee invariance. In the second stage, when using DiT for denoising, can this invariance still be maintained?
- Section 4.4 investigates the performance under different amounts of available training data. To further reflect effect of pretraining in capturing geometric and topological features, could you conduct experiments categorized by structural complexity?

**Limitations:**

yes

**Strengths And Weaknesses:**

- Clear methodological design: This approach first trains a VAE using the PDDFormer encoder to reconstruct atom types, coordinates, and lattice matrices. Subsequently, it employs a DiT to perform flow-matching latent diffusion within the generated latent space. Shifting the diffusion process from a heterogeneous feature space to a latent space is a highly efficient design choice in this domain.
- Significant performance improvements: The model consistently outperforms supervised learning baselines across four JARVIS properties under varying data availability ratios, effectively demonstrating the validity and strength of the latent diffusion pretraining.
- Limited Novelty: The main contribution of this paper lies in applying a latent diffusion model to the latent space learned by a VAE. However, latent diffusion and flow matching in learned latent spaces are well-established techniques. Furthermore, the loss formulation in Section 3.2 appears to be a straightforward adaptation of generic latent flow-matching to this encoder, rather than introducing a theoretic innovation.
- Under-examined invariance: While using PDDFormer to ensure rotational and translation invariances in stage one, processing these features with DiT offers no guarantee that these symmetries are maintained. An experiment is necessary to confirm that the LDM actually preserves the PDDFormer's physical invariances.

---

> ### Author Rebuttal · Authors · 2026-03-31
>
> **Q1. Training Setup and Consistency Across Baselines**
>
> The pretrained baseline methods considered in Table 1 can be broadly categorized into two groups: (i) non-diffusion self-supervised pretraining methods (CrysXPP, CrysGNN, and Crystal Twins), and (ii) diffusion-based pretraining approaches operating in the input space (CrysDiff and DPF). These methods adopt different pretraining objectives (e.g., reconstruction/contrastive learning vs. diffusion-based denoising) and are originally designed with specific encoder architectures. In Table 1, we follow standard practice by using original designs and reported settings of prior pretrain–finetune methods to ensure consistency with published results. As these methods are tied to specific backbones, modifying them requires careful reimplementation. While configurations may differ, results reflect their reported performance.
>
> To provide a controlled comparison and address the reviewer’s concern, we have conducted additional experiments where we replace the encoder backbones with PDDFormer and pretrain all models on the same **large-scale GNoME dataset** used by CrysLDNet, followed by evaluation under identical settings. We note that we cannot report PDDFormer results for CrysDiff, as its fine-tuning source code is not available. Results are reported in the response to **Reviewer tkjV**.
>
> From the tables (see **Reviewer tkjV**), we observe that with the stronger PDDFormer backbone, CrysLDNet achieves average improvements of **4.68%** on JARVIS and **6.44%** on MP over the second-best method, DPF. This analysis allows us to more clearly separate the impact of the pretraining strategy from the choice of backbone, and emphatically shows the effectiveness and novelty of our proposed pre-training method. We thank the reviewer for seeking this clarification.
>
> **Q2. Preservation of Invariance in the Diffusion Stage**
>
> In the **first stage**, the use of PDDFormer as the VAE encoder already ensures rotational and translational invariance in the learned latent space. Since the DiT denoiser operates entirely on these invariant latent representations, the model design inherently preserves these properties. To further validate this, we now conduct an additional experiment where we apply **rotation and translation to the same crystal** and compare the denoised latent outputs. We observe high cosine similarity **(0.985)** and negligible differences **(MAE ≈ 1e-6)**, confirming strong consistency under geometric transformations. These results verify that the latent diffusion stage preserves the invariances established in the **first stage**.
>
> **Q3. Performance Across Structural Complexity**
>
> To address this, we conduct additional experiments by stratifying the training and test crystals into three structural-complexity groups based on the number of atoms (# N) in the unit cell: **simple (#N<=5)**, **moderate (5<#N<10) , and complex (#N=>10)**. For each group, we repeat the limited-data protocol from Section 4.4, i.e., we fine-tune using the same fraction of labeled data within that complexity bucket and evaluate on the corresponding test subset from the same bucket. This evaluates whether pretraining benefits are consistent across different structural complexities, rather than averaged over the full dataset.
>
> We observe a clear monotonic trend in relative improvement, increasing from **2.29% (simple)** to **5.33% (moderate)** and **11.10% (complex)**. This indicates that the benefit of latent diffusion pretraining becomes more pronounced as structural complexity increases. These results indicate the method is particularly effective for complex crystals, capturing richer geometric and topological dependencies where higher-order interactions matter.
>
> |||%|Bandgap(MBJ)|Bulk Modulus|Shear Modulus|SLME|
> |--|--|--|--|--|--|--|
> |# N<=5|PDDFormer|20|0.814|14.21|9.412|10.070|
> ||CrysLDNet|20|**0.790**|**13.85**|**8.851**|**8.761**|
> ||PDDFormer|40|0.701|11.92|8.912|7.636|
> ||CrysLDNet|40|**0.691**|**11.45**|**8.021**|**6.958**|
> ||PDDFormer|60|0.642|11.41|8.721|6.179|
> ||CrysLDNet|60|**0.612**|**10.85**|**7.641**|**5.754**|
> ||PDDFormer|80|0.599|10.85|6.578|5.429|
> ||CrysLDNet|80|**0.565**|**10.14**|**6.201**|**5.228**|
> |# 5<N<10|PDDFormer|20|0.801|14.01|9.012|10.001|
> ||CrysLDNet|20|**0.778**|**13.62**|**8.621**|**7.079**|
> ||PDDFormer|40|0.669|11.88|8.781|7.788|
> ||CrysLDNet|40|**0.653**|**11.25**|**7.891**|**6.311**|
> ||PDDFormer|60|0.609|11.25|8.641|5.851|
> ||CrysLDNet|60|**0.597**|**10.56**|**7.451**|**5.585**|
> ||PDDFormer|80|0.573|10.741|6.352|5.501|
> ||CrysLDNet|80|**0.543**|**10.04**|**5.921**|**4.777**|
> |# N>10|PDDFormer|20|0.789|13.852|8.781|9.901|
> ||CrysLDNet|20|**0.749**|**13.08**|8.297|**5.450**|
> ||PDDFormer|40|0.631|11.01|8.351|6.921|
> ||CrysLDNet|40|**0.606**|**10.72**|**7.567**|**4.658**|
> ||PDDFormer|60|0.598|10.981|8.21|5.820|
> ||CrysLDNet|60|**0.576**|**10.14**|**7.190**|**4.520**|
> ||PDDFormer|80|0.552|10.25|6.214|5.201|
> ||CrysLDNet|80|**0.517**|**9.68**|**5.790**|**4.412**|

---

> > ### Author Rebuttal · Reviewer_cc7e · 2026-04-03
> >
> > The response provided has resolved most of my initial concerns, and I appreciate the additional experiments. While the diffusion process guarantees invariance, I have noticed that the joint training in Stage 2 employs only the unconstrained $\mathcal{L}_{LDM}$. This theoretically risks converging to a trivial global minimum where the latent space loses its expressive capacity. Could the authors provide a principled theoretical analysis explaining why this is avoided, or provide latent variance curves during Stage 2 to demonstrate that this is not occurring? Furthermore, could the authors clarify the rationale for discarding the reconstruction loss during this stage? I would be willing to increase my score if these concerns are addressed.

---

> > > ### Author Response · Authors · 2026-04-05
> > >
> > > **Risk of Latent Space Collapse in Unconstrained Stage-2 Latent Diffusion Training.**
> > > >We thank the reviewer for this insightful follow-up. The concern about latent collapse is valid in principle. We have also discussed this in detail in Appendix E. In our framework, this is mitigated by warm start using  Stage-1 input following the  principle stated in [1]. That is, Stage 2 is initialized from a pretrained VAE (Stage 1), where the latent space is already regularized via reconstruction and KL constraints, ensuring non-degenerate variance. The diffusion objective then refines this space by encouraging smooth, predictable latent trajectories, which implicitly maximizes mutual information between noisy and clean latents.
> > >
> > > > We have also done a detailed ablation study in Appendix **(Check Fig 5)** Here we compare outcome of  joint training with and without the pretrained VAE warm start, the loss collapses almost immediately to the order of $10^{−3} ⁣− ⁣10^{−4}$, whereas CrysLDNet remains stable under joint VAE–Flow training.
> > > >
> > >
> > > > Further as per the suggestion of the reviewer, to directly verify that Stage 2 does not drive the encoder toward a trivial representation,  we additionally track the mean variance of node-level latent embeddings during Stage 2 on a fixed validation batch (Fig. [Latent Variance Stability](https://anonymous.4open.science/r/Latent_Varience_Stability-2BD5/latent_variance.png)). As shown in the attached figure, after a brief initial transient the latent variance remains stable and clearly non-zero **(roughly 1.9 ⁣− ⁣2.05)** across epochs, indicating that the latent space retains expressive capacity rather than collapsing to a constant solution.
> > >
> > > **Latent Variance Stability Figure** :- https://anonymous.4open.science/r/Latent_Varience_Stability-2BD5/latent_variance.png
> > >
> > > [1] Chien, Jen-Tzung, and Tien-Ching Luo. "Flow-Based Variational Sequence Autoencoder." 2022 Asia-Pacific Signal and Information Processing Association Annual Summit and Conference (APSIPA ASC). IEEE, 2022.
> > >
> > > **Furthermore, could the authors clarify the rationale for discarding the reconstruction loss during this stage?**
> > >
> > > >The objectives of the two stages are fundamentally different. In Stage 1, the VAE encoder learns a smooth and compact latent representation of 3D crystal structures via reconstruction loss, which enforces local, structure-level fidelity. In Stage 2, the goal shifts to refining these representations using latent diffusion, which enforces distribution-level consistency and promotes a smooth, invariant manifold. Retaining reconstruction at this stage may introduce conflicting optimization signals—reconstruction favors exact input recovery, while diffusion encourages flexibility and global structure modeling. This can overly constrain the latent space and hinder learning of task-relevant features.
> > > >To validate this design, we conduct an **ablation where reconstruction is retained in Stage 2 (LDM+Decoder)**. This variant consistently underperforms CrysLDNet on JARVIS, demonstrating that removing reconstruction enables better representation learning and improved downstream performance.
> > >
> > > |Method|Formation Energy|Band Gap (OPT)|Total Energy|Ehull|Bandgap (MBJ)|Bulk Modulus|Shear Modulus|SLME (%)|Spillage|
> > > |------|----------------|---------------|-------------|------|--------------|--------------|----------------|-----------|----------|
> > > |LDM+Decoder|0.034|0.135|0.032|0.054 |0.286|10.38|9.673|4.641|0.365|
> > > |CrysLDNet|**0.026**|**0.118**|**0.027**|**0.032**|**0.242**|**8.817**|**8.528**|**4.256**|**0.340**|

---

### Official Review · Reviewer_8afe · 2026-03-15

**Soundness:** 2
**Presentation:** 3
**Significance:** 2
**Originality:** 1
**Overall Recommendation:** 4
**Confidence:** 3

**Summary:**

This paper studies crystal property prediction under limited labeled data. The authors propose a latent diffusion pretraining framework (CrysLDNet) that leverages large collections of unlabeled crystal structures to learn structural representations before fine-tuning on downstream property prediction tasks.
The approach first encodes crystal structures into a latent representation using a variational autoencoder (VAE). A diffusion model is then applied in the latent space to refine these representations during pretraining. After pretraining, the model is fine-tuned to predict target material properties. The authors demonstrate that the pretrained representations improve downstream prediction accuracy, particularly in low-data regimes.
Overall, the paper explores an interesting direction of using latent diffusion models for representation learning in materials science.

**Compliance With Llm Reviewing Policy:**

Affirmed.

**Final Justification:**

As few of my concern has been justified, let me increase the score from 3 to 4.

**Key Questions For Authors:**

- How does the proposed approach compare with other self-supervised pretraining strategies that do not use diffusion models?
- What specific advantage does latent diffusion provide compared to simpler representation learning objectives?
- Can the authors provide additional ablations isolating the contribution of the diffusion component relative to the VAE encoder?

**Limitations:**

yes

**Strengths And Weaknesses:**

Strengths

- The paper studies an important problem in machine learning for materials science. Crystal property prediction often suffers from limited labeled data, making representation learning approaches potentially valuable.
- The motivation for leveraging large collections of unlabeled crystal structures is reasonable and aligns with recent trends in self-supervised learning.
- The framework is relatively flexible and can be combined with different backbone encoders, which may make it broadly applicable.
- Experimental results suggest consistent improvements over purely supervised baselines, particularly in low-data settings.

Weaknesses

- While the motivation is well articulated, the methodological novelty appears somewhat limited. The proposed framework largely follows a standard pretrain–finetune paradigm with a VAE encoder and diffusion-based latent refinement.
- The key idea of performing diffusion in latent space rather than input space has been explored in other domains. The contribution here appears to be primarily an application of this idea to crystal representations.
- The paper does not clearly demonstrate why latent diffusion specifically is necessary for the task, compared to simpler self-supervised pretraining methods.
- Some components of the framework appear relatively modular and incremental rather than introducing fundamentally new modeling ideas.
- The empirical improvements are positive but moderate, and it is somewhat unclear whether they justify the additional complexity of the pretraining pipeline.

---

> ### Author Rebuttal · Authors · 2026-03-31
>
> **Regarding limited novelty**
>
> We respectfully disagree with the reviewer’s assessment of limited novelty. Novelty can arise not only from new architectures but also from principled adaptation of existing methods to underexplored problems. While latent diffusion is widely used for generative tasks in vision and molecules, its use as a pretraining strategy for representation learning in crystal property prediction remains underexplored. Our work introduces this direction by adapting latent diffusion for this purpose.
>
> - **Latent diffusion for pretraining (not generation)**:
>  > Unlike prior works that use latent diffusion for generative modeling, we employ it as a self-supervised pretraining objective to learn transferable representations for property prediction. To the best of our knowledge, this direction has been minimally explored, especially in the context of crystalline materials.
> - **First application in crystal pretraining (to our knowledge)**:
>  > Existing crystal pretraining methods primarily rely on self-supervised objectives or feature-space diffusion. Prior diffusion-based approaches for crystals operate directly in the input space. In contrast, we are the first (to the best of our knowledge) to introduce latent diffusion–based pretraining for crystal property prediction, providing a new paradigm for representation learning in this domain.
> - **Addressing domain-specific challenges**:
> > Crystal structures are inherently heterogeneous, comprising discrete atom types, continuous lattice parameters, and periodic coordinates, making feature-space diffusion complex. Our approach addresses this by learning a unified, smooth latent space via a symmetry-preserving encoder, enabling more effective diffusion.
> - **Physically meaningful latent space**:
> > The latent space preserves key crystal symmetries (rotation, translation, periodicity), ensuring physically consistent representations. This leads to stronger reconstruction and downstream gains (e.g., **~10%** improvements in atom/lattice reconstruction), highlighting the importance of symmetry-aware latent diffusion.
>
> We will revise the manuscript to better position our work and clarify its novelty for crystal property prediction.
>
> **Q1. Comparison with non-diffusion pretraining methods**
>
> We note that this comparison is already discussed in the paper. We analyze non-diffusion self-supervised methods (e.g., CrysXPP, Crystal Twins, CrysGNN) in the Introduction (paras. 2–3), with further details in Appendix D.2. We also provide empirical comparisons in Table 1 and Section 4.2, where CrysLDNet consistently outperforms these methods. In particular, Table 1 shows improvements of **36.2%**, **38.7%**, and **50.1%** over CrysXPP, Crystal Twins, and CrysGNN, respectively. Additionally, we conduct experiments using a stronger PDDFormer backbone with unified GNoME pretraining (see Reviewer **tkjV**). Under this setting, all methods improve and the gap reduces; however, CrysLDNet still achieves gains of **16.8%**, **15.4%**, and **12.4%**, confirming its **effectiveness beyond backbone and data scale**.
>
> **Q2. Specific advantages of latent diffusion**
>
> Compared to simpler objectives (e.g., reconstruction, contrastive learning, one-step denoising), diffusion-based pretraining provides a richer learning process via multi-step denoising, capturing both local and global structure. This improves representation quality and generalization (Secs. 4.2–4.4), especially in low-data regimes; e.g., at **40% training data**, CrysLDNet achieves average improvements of **22.2%** and **17.9%** over CrysGNN and CrysDiff.
>
> Beyond this, diffusion in a learned latent space avoids input heterogeneity (e.g., atom types and periodic coordinates), yielding compact, smooth representations that better preserve structure. As shown in **Sec. 4.5**, this improves reconstruction (atoms, coordinates, lattice) and downstream performance vs. simpler objectives and input-space diffusion methods **(Table 1, Sec. 4.2)**.
>
> **Q3. Isolating the effect of the diffusion component**
>
> We note that we have already conducted detailed ablation studies to isolate the contribution of the diffusion component relative to the VAE encoder. In **Section 4.7 (Table 4)**, we explicitly compare three settings: VAE-only, LDM-only, and the full CrysLDNet (VAE+LDM).
>
> The results show that:
> - Both VAE-only and LDM-only setups lead to consistent performance degradation across all evaluated properties.
> - The full model (VAE+LDM) achieves the best performance, demonstrating that the two components are complementary and jointly necessary.
> - Notably, LDM-only performs better than VAE-only, indicating the strong role of diffusion in learning richer representations.
>
> Together, these results clearly demonstrate that while the VAE provides a meaningful initialization, the diffusion component plays a critical role in enhancing representation quality, and the best performance is achieved when both are used jointly.

---

### Decision · Program_Chairs · 2026-04-30

**Decision:**

Accept (regular)

**Comment:**

In this submission, the authors proposed a latent-diffusion pretraining method for crystal property prediction, mitigating the data scarcity issue. The reviewers acknowledged the contribution of this work and leaned towards weak acceptance, and AC agreed with this decision.